# FACILITATING CAUSAL STUDIES ON THE LEARNABILITY OF FORMAL LANGUAGES

## ABSTRACT

A common approach to studying the learnability of neural language models is to use formal languages. We build on this and introduce a controlled sampling procedure for probabilistic finite-state automata. Our method enables count-based interventions on the generative process: we can directly generate corpora with an exact number of occurrences of targeted properties—such as symbols or states. The approach efficiently samples corpora under interventions, enabling causal studies. Specifically, it allows us to ask how the salience of the properties we target causally impacts the learnability of language models. We experimentally validate the efficiency of the sampling in two studies. We first analyze how local properties of automata predict the learnability of transitions associated with the properties. We then show how causally intervening on the number of times a property, such as the occurrence of a given symbol, results in different learnability than if the training set was gotten with ancestral sampling. Our findings indicate that using the standard sampling method overestimates the effect of training on fewer occurrences, while the importance is underestimated for higher occurrence counts. In doing so we demonstrate how to efficiently conduct causal studies of language models' learnability of formal languages.

## 1 INTRODUCTION

Finite-state automata (FSA) have proven themselves a useful tool for studying how neural language models (LMs) learn. Existing work has leveraged manually constructed ones to study particular phenomena that are difficult for neural LMs to learn (Lake & Baroni, 2018; Ruis et al., 2020; Hupkes et al., 2020; Allen-Zhu & Li, 2024). Recently, sampling random automata that span an entire language class has proven particularly attractive (Valvoda et al., 2022; Borenstein et al., 2024). The appeal of randomly sampled formal languages comes from the fact that they allow the researcher to generate infinitely many strings in a highly controlled way—enabling evaluation over entire classes of languages as opposed to single instances. Sampling from formal languages allows us to correlate properties of language with the performance of different architectures (e.g., Linzen et al., 2016; Jawahar et al., 2019; Liu et al., 2019; Manning et al., 2020; Rogers et al., 2021). If one is, however, interested in a causal analysis(Elazar et al., 2023; Chen et al., 2024)—for example, to what degree specific features of a language causally impact its learnability—one would need to causally intervene on the generative process behind the language (Pearl et al., 2016). This has only been possible by manually crafting such interventions. In this work, we introduce a methodology that enables such experiments at scale through controllable causal interventions on the process that produces the language.

Causally intervening on a process that produces a language is not straightforward, however. One could somehow try to manually modify a given automaton. But manually changing a machine would only result in a corpus that allows us to ask, what if this structural property were different? This would not allow us to ask higher-level questions such as: *What if we had a corpus of size K sampled from a given machine, with N occurrences of a target symbol?* Another approach would be to use rejection sampling, iteratively sampling from a machine that defines a language, and throwing away those samples that do not meet a given criteria. In practice, this would often be prohibitively slow—the samples we are looking for might well have low-probability.

We solve this problem by introducing a new algebraic structure for sampling from formal languages that can be defined by probabilistic weighted automata—the **marginal semiring**. The marginal

semiring allows us to track the number of occurrences of pre-determined events such as symbol, transition, and state occurrences when sampling from any probabilistic finite-state automaton (PFSA). This facilitates algorithms for controlled counting-based sampling, where we can *condition* on the properties we would like our datasets to have. We develop a two-step approach to sampling a corpus under occurrence constraints: First, we sample how often a given property should occur in each sampled string. Then, we sample each string under the constraint that the property occurs that often. For instance, we might want to see 100 occurrences of the symbol 'a' in 1000 strings.

The methods we present for sampling from PFSAs under intervention are both efficient and applicable to many types of interventions. Our approach is also asymptotically faster than naïve implementations using rejection sampling. In our experiments, we demonstrate how one can use counting interventions to study how often the PFSA makes use of specific transitions, states or symbols to generate strings. We can the study how these interventions impact the learnability of formal languages by neural language models. We train Transformer and LSTM language models and find Transformers to generally perform better when measured using targeted KL-divergence. The Transformers consistently perform better when all target features are held out, but the LSTMs benefit more from additional examples. More significantly, our interventions allow us to concretely single out the significant differences in what local and global properties of the automata predict the performance of the two architectures. For example, we find that Transformers are more sensitive to the properties of source states of the transitions we intervene on, and the LSTMs more so to the transition target states. We finally conduct a more direct causal study where we ask: how does the number of occurrences of a given symbol impact its learnability? We use Monte Carlo sampling to approximate the expected decomposed KL-divergence for the target symbol, finding that standard sampling methods overestimate the effect of training on fewer occurrences, while the importance is underestimated for higher occurrence counts. While ancestral sampling gives a linear trend, causal sampling results in an exponentially decaying trend—highlighting why causal studies are important, those based on mere correlations are by no means guaranteed to capture the causal relationship we wish to explore.

## 2 CAUSAL GRAPHICAL MODELS FOR SAMPLING FROM AUTOMATA

We now develop a methodology for sampling from LMs under count-based interventions on properties that can be described as sets of transitions. An SCM is a directed graph whose nodes represent variables and whose arrows represent causal relationships between them. Unlike in general graphical models, where the topology of the graph describes conditional (in)dependencies, the edges in an SCM indicate *causal* relationships—changing variables causally influences the variables downstream. The causal nature of the model allows us to define interventions, which intuitively manifest themselves as modifications of the causal graph: An intervention on a node $X$ removes the causal dependence on its parent nodes, allowing us to isolate the downstream effect of that particular intervention. We use the do-operator to indicate the effect on downstream nodes $Y$ with $P(Y \mid \text{do}(X = x))$ (Pearl et al., 2016).

We apply the SCM framework to causally intervene on the datasets sampled from a finite-state automaton $\mathcal{A}$ (see App. C for a formal definition of automata). To do so, we define an SCM in which we are able to intervene on properties of interest and conditionally sample strings based on the interventions. The SCM that models *property occurrences* is presented in Fig. 1. It contains the following random variables (RVs) $\mathcal{A}$ over the number of machines our configurations allow, the number of times the target property is seen $P$, and the size of the dataset $K$. We also use $\mathcal{N}$ to denote the trained neural model RV and the samples used to train it are indicated by $\sigma_k$, and the automata paths corresponding to those as $\Pi_k$. We use a triangular shape to indicate a deterministic random variable, and the factor notation (Kschischang et al., 2001) to indicate that the RVs $P$ and $\Pi_k$ are jointly distributed without being specific about their relationship.

Given a property $P$, our interventions take then the form $\text{do}(K = k, P = n)$ for some constants $k \in \mathbb{N}$ and $n \in \mathbb{N}$. In the next section, we describe how we can construct automata that enable controlled interventions via these RVs.

## 3 THE MARGINAL SEMIRING

We wish to intervene on the language-generating process of PFSAs (Def. C.3) to generate corpora under different interventions. Given a PFSA $\mathcal{A}$ and some constraint $\phi$, this corresponds to sampling strings $\sigma$ from the posterior distribution $p_{\mathcal{A}}(\sigma \mid \phi)$. Concretely, we want to control the number of times that particular transitions in $\mathcal{A}$ are taken when a corpus is sampled.

To perform such experiments at scale, it is vital that sampling from $p_{\mathcal{A}}(\sigma \mid \phi)$ be done efficiently. An essential contribution of this work is a new algorithm that is asymptotically faster than naïve sampling approaches. Take symbol interventions, for instance—suppose we want to sample strings with exactly $k$ occurrences of symbol $a$, where $0 \leqslant k \leqslant K$. We could easily—separately for each $k$—construct a PFSA encoding the language of strings that contain precisely $k$ $a$'s, intersect it with the PFSA encoding our language of interest, push the weights to make it probabilistic, and sample from the resulting PFSA. This approach would take $\mathcal{O}(|Q|^3 n^4)$ time, where $|Q|$ is the number of states and $n$ is the maximum count targeted. In this section, we provide an asymptotically faster method than this approach in $\mathcal{O}(|Q|^3 n^2)$ time, which goes down to $\mathcal{O}(|Q|^3 n \log n)$ by making use of the fast Fourier-transform (App. E.1). A more detailed of the runtime is given in App. E.2.

The key idea is to not perform an intersection separately for each $k$, but to redefine the weights of the PFSA not as probabilities, but as *vectors* of size $K + 1$, so that applying weight pushing *once* computes the posterior distribution for all $k$ at once. We propose a new semiring, which we call the **marginal semiring**, that facilitates this. First, we will give a formal definition of semiring—for a more detailed definition, and a definition of a *monoid*, see App. B.

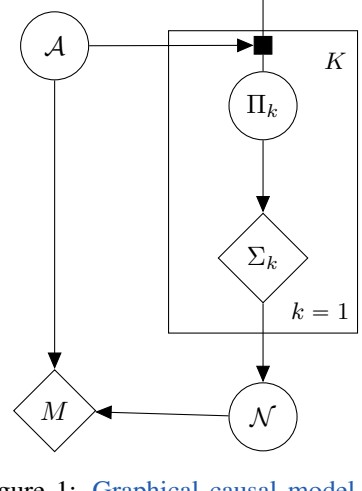

Figure 1: Graphical causal model for evaluating the effect of interventions on a property $P$, as measured by the effect measure $M$ to compare the trained model $\mathcal{N}$ and the automaton $\mathcal{A}$. $\Pi_k$ is a given path, and $\Sigma_k$ the string the corresponds to it.

**Definition 3.1** (Semiring). *A **semiring** is a quintuple $(\mathbb{K}, \oplus, \otimes, \mathbf{0}, \mathbf{1})$ where (i) $(\mathbb{K}, \oplus, \mathbf{0})$ is a commutative monoid with identity element $\mathbf{0}$, (ii) $(\mathbb{K}, \otimes, \mathbf{1})$ is a monoid with identity element $\mathbf{1}$, (iii) multiplication left and right distributes over addition: $a \otimes (b \oplus c) = (a \otimes b) \oplus (a \otimes c)$ and $(b \oplus c) \otimes a = (b \otimes a) \oplus (c \otimes a)$, and (iv) Multiplication with $\mathbf{0}$ annihilates $\mathbb{K}$: $\mathbf{0} \otimes a = a \otimes \mathbf{0} = \mathbf{0}$. Furthermore, let $a^{\otimes i} = \bigotimes_{j=1}^{i} a$, and let $a^* = \bigoplus_{i=0}^{\infty} a^{\otimes i}$. If $a^*$ is defined and in $\mathbb{K}$ for all $a \in \mathbb{K}$, we say the semiring is **closed**. In that case, $a^* = \mathbf{1} \oplus a \otimes a^* = \mathbf{1} \oplus a^* \otimes a$.*

**Using semirings to count occurrences.** Let us first discuss the intuition of the marginal semiring before formally defining it below. Given a PFSA, we construct a related FSA with weights in $\mathbb{R}^{K+1}$. For each weight $\mathbf{v}$, the element $\mathbf{v}_i$ is the probability of sampling exactly $i$ occurrences of our target feature. This feature could, for instance, be a symbol $a$: If a transition with probability $w$ emits or scans the symbol $a$, then we map it to the weight $[0, w, 0, \ldots, 0]$, indicating that a single occurrence of the symbol $a$ appears with probability $w$. If the transition scans anything other than $a$, we map it to the weight $[w, 0, \ldots, 0]$, indicating that zero occurrences of $a$ appear with this probability. In a semiring-weighted FSA, the weights of transitions along a path are combined multiplicatively. To reflect that, we want the multiplication of two weights in the marginal semiring to shift the probability $w$ to the position that is the *sum* of the prior two positions, which corresponds to the occurrence of the number of symbols equal to the length of the path. The weight for a single path always has, at most, one non-zero entry. We also wish to be able to aggregate multiple paths together, something we do with elementwise addition; in this case, entry $i$ still captures the total probability of sampling $i$ occurrences. We can compute the backward weights of every state in the automaton to get a vector of such probabilities. We then apply a sampling procedure that, starting at the start state, initializes a counter $i$ to the target $k$, then uses probability distributions based on entries at $i$ for sampling, and decrements $i$ whenever a transition emits $a$.

We now define this notion of the marginal semiring formally. Note that we generalize this so that it can be applied not only to a PFSA with probabilistic weights in $\mathbb{R}$ but with any base semiring.

**Definition 3.2.** *(Marginal semiring) Let $(\mathbb{K}, +, \times, 0, 1, \star)$ be a closed semiring, and let $N \in \mathbb{N}$. We refer to this semiring as the **base semiring**. The $N^{th}$-order **marginal semiring** with respect to $(\mathbb{K}, +, \times, 0, 1, \star)$ is the sextuple $(\mathbb{K}^{N+1}, \oplus, \otimes, \mathbf{0}, \mathbf{1}, *)$, such that for all $\mathbf{v}, \mathbf{v}' \in \mathbb{K}^{N+1}$ and $0 \leqslant i \leqslant N + 1$: (i) $(\mathbf{v} \oplus \mathbf{v}')_i \stackrel{\text{def}}{=} \mathbf{v}_i + \mathbf{v}'_i$ (ii) $(\mathbf{v} \otimes \mathbf{v}')_i \stackrel{\text{def}}{=} \sum_{n=0}^i \mathbf{v}_n \times \mathbf{v}'_{i-n}$ (iii) $\mathbf{0} \stackrel{\text{def}}{=} [0, 0, \ldots, 0]^\top$ (iv) $\mathbf{1} \stackrel{\text{def}}{=} [1, 0, \ldots, 0]^\top$ (v) $\mathbf{v}^* \stackrel{\text{def}}{=} \bigoplus_{n=0}^\infty \mathbf{v}^{\otimes n} = \bigoplus_{n=0}^\infty \underbrace{\mathbf{v} \otimes \cdots \otimes \mathbf{v}}_{n \text{ times}}.$*

We note that the marginal semiring of degree $N$ is isomorphic to a truncated polynomial semiring, i.e. the quotient ring over the ideal of polynomials of degree $N + 1$. We derive this in App. G.

The multiplication operation, a convolution, gives us the counting property we describe intuitively above. We show that the marginal semiring satisfies the semiring axioms in App. D. We derive a closed-form solution to the star operator in App. E that enables an efficient algorithm for calculating the path sums between any two nodes in a PFSA defined over the marginal semiring. Furthermore, as described in App. E.1, we implement $\otimes$ using the fast Fourier transform (FFT) if the base semiring is the real semiring. We now demonstrate how the marginal semiring can be used for counting sets of transitions in a PFSA.

**Definition 3.3** (Marginal automaton). *Let $\mathcal{A}$ be a PFSA and $\phi$ a **feature function** $\phi \colon \Delta \to \{0, 1\}$, where $\Delta$ is the set of transitions in $\mathcal{A}$. The transitions that are assigned to 1 are those that we wish to count. We define the **lifting function** $\mathcal{L}_\phi \colon \mathbb{K} \to \mathbb{K}^{N+1}$ as $\mathcal{L}_\phi(\alpha)_i = \alpha \cdot \mathbb{I}[i = \phi(\alpha)]$. The lifting function maps the weights from $\mathcal{A}$ to the new **marginal automaton**, denoted by $\mathcal{A}_{\mathcal{L}_\phi}$.*

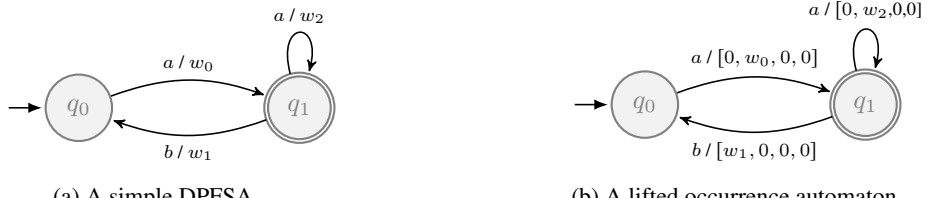

(a) A simple DPFSA.      (b) A lifted occurrence automaton.

Figure 2: The lifted occurrence automaton is on the right, with the original automaton to the left. We target the symbol $a$ for four occurrences. $q_0$ is the starting state and $q_1$ the accepting state.

We provide a simple example of the marginal semiring and automaton in Fig. 2. We see on the right how the weights for individual transitions have been modified when we lift the symbol $a$, the weights for $a$ in the original automaton have been put in the second place on the vector, while the weight for $b$ is in the first place. Let's now consider the string "aaba" as a concrete example, in the left-hand side automaton defined over the real semiring, the probability of the string is given by $w_0 \cdot w_2 \cdot w_1 \cdot w_0$. In the right-hand semiring multiplication is defined as a convolution ($\otimes$), we thus get the path weight $[0, w_0, 0, 0] \otimes [0, w_2, 0, 0] \otimes [w_1, 0, 0, 0] \otimes [0, w_0, 0, 0] = [0, 0, w_0 \cdot w_2, 0] \otimes [w_1, 0, 0, 0] \otimes [0, w_0, 0, 0] = [0, 0, 0, w_0 \cdot w_2 \cdot w_1, 0] \otimes [0, w_0, 0, 0] = [0, 0, 0, w_0 \cdot w_2 \cdot w_1 \cdot w_0]$ We see that whenever the symbol we target is seen ($w_0$ and $w_2$), the original path weight moves up an index in the occurrence semiring weight, this allows us to read the non-zero index to count the number of times the symbol was seen.

We define the commonly used terms **path**, **path weight**, and **backward weight** in App. C.1. We now formally derive the marginal automaton behavior, which we explained intuitively earlier. We first prove the intuition that the weight for an individual run is a one-hot vector whose position encodes the desired number of occurrences.

**Theorem 3.1** (Path Weight Interpretation). *Let $\phi$ be a feature function and $\mathcal{A}_{\mathcal{L}_\phi}$ its marginal automaton. We denote the number of times a feature occurs on a path $\boldsymbol{\pi}$ as $|\boldsymbol{\pi}|_\phi$. If $\boldsymbol{\pi}$ is a path in $\mathcal{A}$, and $w_I(\boldsymbol{\pi})$ is the path weight, the following holds:*

$$|\boldsymbol{\pi}|_\phi = \operatorname*{argmax}_{1 \leqslant i \leqslant N} w_I(\boldsymbol{\pi})_i \text{ and } \forall j \neq |\boldsymbol{\pi}|_\phi, w_I(\boldsymbol{\pi})_j = 0 \tag{1}$$

*In words, the index of the only non-zero element of $\mathrm{w}_I\left(\boldsymbol{\pi}\right)$ tells us how often the feature occurs in $\boldsymbol{\pi}$.*

*Proof.* See App. F ∎

The following theorem formalizes the intuition that aggregating run weights by summing them element-wise results in vectors that encode the weights of sampling specific numbers of occurrences.

**Theorem 3.2** (Pathsum Interpretation). *Let $\mathcal{A}$ be a PFSA and $\Pi$ be a random variable over the paths in $\mathcal{A}_{\mathcal{L}_\phi}$. Then, $|\Pi|_\phi$ is also a random variable and we have*

$$p(|\Pi|_\phi = n) = \boldsymbol{\beta}_{\mathcal{A}_{\mathcal{L}_\phi}}(q)_n. \tag{2}$$

*In words, the probability of exactly $n$ occurrences of the feature in the string scanned by a randomly sampled path is the $n$-th element of the backward weight for $n \in \{0, 1, \ldots, N\}$.*

*Proof.* See App. F. ∎

## 4 SAMPLING UNDER FEATURE CONSTRAINTS

We now use the marginal automaton to develop tools for sampling under feature-counting interventions. Let $\mathcal{A}$ be a PFSA. We wish to sample $K$ strings with a total of $N$ occurrences of the features $\Phi$ satisfying some feature function $\phi$. First, we must sample how often the features should appear in a given string.

**Theorem 4.1** (Probability over set of strings). *Let $(\mathrm{K}_i)_{i \in I}$ be a set of indexed strings sampled from a PFSA, and $|(\mathrm{K}_i)_{i \in I}|_\phi$ denote the number of occurrences of the feature in all strings combined. The probability of seeing $n$ occurrences from $\phi$ in $(\mathrm{K}_i)_{i \in I}$ is given by*

$$P(|(\mathrm{K}_i)_{i \in I}|_\phi = n) = (Z^{\otimes k})_n, \tag{3}$$

*where $k = |I|$ and $Z$ is the pathsum of the marginal semiring acquired by lifting the automaton while targeting the features.*

*Proof.* See App. F. ∎

**Theorem 4.2** (Sampling lengths). *Let $Z \in \mathbb{R}^{N+1}$ be the pathsum of the lifted marginal automaton $\mathcal{A}_{\mathcal{L}_\phi}$, corresponding to some PFSA we wish to sample from, for some target features $\phi$. Let $\mathrm{K}_k$ be the $k$-th string sampled. Assuming that we have assigned $m$ out of $N$ symbols to the first $k-1$ sampled strings, then the probability of seeing $n$ symbols in the next string is given by*

$$p(|\mathrm{K}_k|_\phi = n) = Z_n \cdot (Z^{\otimes K-k-1})_{N-m-n}. \tag{4}$$

*Proof.* See App. F. ∎

Thm. 4.2 tells us how many features we should ask for in each string when sampling under intervention. We have now presented the necessary background to state how to sample from a lifted machine $\mathcal{A}_{\mathcal{L}_\phi}$. If we have already observed $n$ of the $N$ desired features in the last $k$ strings, then we sample using the following corollary.

**Corollary 4.1.** *(Symbol Occurrence Sampling) Sampling from $\mathcal{A}_{\mathcal{L}_\phi}$, using the following procedure results in a string where the expected number of occurrences of the target feature is $N$.*

$$p(q \xrightarrow{\delta} q') \propto (\mathbf{v} \otimes \beta(q'))_{N-(n+\phi(\delta))} \tag{5}$$

*Here, $n$ is the number of times we have observed the target feature and $\mathbf{v}$ is the weight of the transition.*

To summarize, Thm. 4.2 and Cor. 4.1 tell us how to sample from $\mathcal{A}_{\mathcal{L}_\phi}$ so that we get a specific number of expected target features in a corpus of a fixed size. For each sampled string, we first sample how often we should see the feature using Thm. 4.2, and then we proceed to sample using Cor. 4.1. The following section demonstrates how this can be applied in practice.

## 5 EXPERIMENTAL SETUP

We use the marginal semiring as a tool to study how causally intervening on symbols, transitions, and states affects a neural model's ability to learn regular languages defined by PFSAs. Based on this, we can begin to analyze what properties of a language are more challenging for a neural LM to learn. Our approach is straightforward. We first sample a large number of PDFAs $(\mathcal{A}_m)_{m=1}^{M}$. From each $\mathcal{A}_m$, we sample $K$ strings $\overline{\sigma} = \{\overline{\sigma}_n\}_{n=1}^{K}$ with $N$ occurrences of the target features $\phi$. For each $\overline{\sigma}$ we then train a neural language model and evaluate its ability to learn the weighted language of the *original* PDFA.

### 5.1 PROPERTY INTERVENTIONS

We investigate three kinds of causal interventions: on the number of times a certain **transition**, **state**, or **symbol** is seen during training. Our three types of interventions are best described with do-notation introduced in §2. Each of these is captured by some property $P$, by intervening on it we sample according to

$$\overline{\sigma} \sim p_{\mathcal{A}}(\cdot \mid \mathrm{do}(K = k, P = n)) \tag{6}$$

### 5.2 SAMPLING PDFAS

We sample random PDFAs, with 100 states over an alphabet of 10 symbols. The sampling procedure of a single automaton $\mathcal{A}$ is as follows: For each source state $q \in Q$, we sample a set of symbols, $y \in \Sigma$, where each symbol has a 0.5 chance of being included. We randomly sample a target state $q' \in Q$ for each symbol. This gives us a set of unweighted transitions between states and associated symbols. We set each state to be accepting with a probability of 0.1. Finally, for each state, we use Dirichlet sampling (see App. I) to randomly sample the probabilities for the outgoing transitions and the acceptance. In total, we sample 74 machines for the transition interventions, 149 machines for the state interventions, and 73 machines for the symbol interventions.[1] Note that we train more than a dozen neural networks for each sampled machine.

The configuration of the neural language models, including specific hyper-parameters is given in App. J.

### 5.3 KL-DIVERGENCE BETWEEN PFSAS AND TRAINED MODELS

To evaluate the performance of the trained models against the sampled automata, we use the Kullback–Leibler divergence between the trained model and the PFSA from which the training data was sampled. That is, $\mathrm{KL}(p_{\mathcal{A}} \| p_{\theta}) = \Sigma_{x \in \Sigma*} p_{\mathcal{A}}(x) \log \frac{p_{\mathcal{A}}(x)}{p_{\theta}(x)}$, where $p$ is the probability mass function of a string according to a PFSA $\mathcal{A}$, and $p_{\theta}$ represents the probabilities of the trained model. This well-known measure captures how different the two distributions are.

**Decomposed KL.** We also introduce a decomposed KL divergence to evaluate the effect of the interventions. We evaluate these empirically by sampling a held-out corpus $\overline{\sigma}_{\mathrm{test}} = \{\overline{\sigma}_n\}_{n=1}^{K}$ for $\mathcal{A}$ while keeping track of what transitions were used, giving us a sequence $(\sigma_i, q, w, q', (\sigma_j)_{j<i}) \in \Sigma \times Q \times \mathbb{R} \times Q \times \Sigma^{i-1}$. We overload $\overline{\sigma}_{\mathrm{test}}$ for brevity to also refer to these tuples. Let $\pi_{\mathcal{A}}(q)$ be the forward probability (i.e., the sum of all path weights for paths ending in the state) of being in state $q$. We can then express the KL-divergence over $\overline{\sigma}_{\mathrm{test}}$ as follows:

$$\mathrm{KL}_{\mathrm{E}}(p_{\mathcal{A}} \| p_{\theta}) = \sum_{\overline{\sigma}_{\mathrm{test}}} \pi_{\mathcal{A}}(q) \cdot \log \frac{w}{p_{\theta}(\sigma_i | \sigma_{j<i})}. \tag{7}$$

We can then constrain this decomposition to exact transitions relevant to our three interventions. We simply limit the samples we marginalize over: If we target a symbol, we only include the data points containing the target symbol in the first position. If we target a single transition, we only include the

---

[1]The variations in the number of machines are not something we planned for but rather an artifact of how long we were able to keep the processes running.

entries corresponding to that transition, i.e., where the symbol, source, and target state are those we are interested in. For state interventions, we only consider the elements where the target state is the intervention state. We report results as the average divergences over the held-out samples. A more in-depth treatment of the decomposed KL is given in App. H.

## 5.4 SECOND ORDER ANALYSIS

Our goal is to understand which automata properties can explain the benefit of seeing more occurrences of the intervention targets. We first fit linear models to the trends for each machine, giving us a collection of linear models, each with two coefficients. These coefficients encode the specific trend for a given machine. Then the natural question is: Can we explain the difference in these coefficients using properties of the sampled machines? We do so by conducting a second-order analysis of the coefficients of the fitted curves.

Specifically, the above-mentioned linear models are fitted using an ordinary least-squares linear model (OLS), predicting each automaton's KL values given the occurrence count. The OLS models are given by $y = \alpha_0 + \alpha_1 x$, where $x$ is the occurrence count for a given automaton, and $y$ is the KL we target. The second-order analysis is then fitted using a weighted least-squares model (WLS). We do this for the intercepts ($\alpha_0$), to get the baseline values for zero occurrences, and the slopes ($\alpha_1$). The WLS model is given by $y = \beta_0 + (\beta_1 + \ldots + \beta_{n-1})x$ where the $\beta$'s are the explanatory variables we list below, and the $x$s are the $\alpha$s from the OLS models, and $y$ corresponds to the KL values. The WLS model is trained with a weighted maximum likelihood objective $\sum w_i (y_i - y)^2$, where the weights are the inverse squared standard error of the $\alpha$s, to account for their uncertainty. Details of the explanatory variables we consider are given in App. K.

## 6 COMPARING LOCAL LEARNABILITY OF TRANSFORMERS AND LSTMS

We now consider the intervention categories one at a time and ask what explanatory variables could explain the trends we see, relying on our second-order analysis (see §5.4). In all intervention categories, we find that the Transformers perform better than the LSTM RNNs, and that the RNNs benefit double or more from increased occurrences. You can find our full results in App. L, we now briefly discuss key findings from the second-order analysis.

**Transition Interventions.** We first find that the intercepts (zero occurrences of a transition) are higher for the RNNs than the Transformers, yet the RNNs benefit almost twice as much from an increased number of target occurrences—a recurring pattern for all intervention categories. Second, we observe that the source entropy path-sum, which encodes the complexity of reaching a specific state has the strongest effect on the intercepts. This is perhaps unsurprising, as a higher entropy indicates more variation in the strings leading up to the target transition. Finally, we find, that the Transformers are more sensitive to the source entropies in modeling the slopes, while the RNNs respond to both the source and target entropies. The specifics of the modeling results are given in Tab. 2 and Tab. 5 in the Appendix. Examples of the trends we got from transition interventions are given shown in Fig. 3b.

**State Interventions.** We observe a similar pattern of the state intervention intercepts like above, the entropy path sum at the state is the most influential predictor. Oddly enough, it has a small but significant positive trend for the decomposed KL. Increasing the number of occurrences also leads to lower divergence, with the slope being negative and significant. For the slope, the local entropy is again hindering the Transformers, while the forward entropy is a more limiting factor for the RNNs. This hints at a fundamental difference in how the two architectures solve the problem of predicting the next symbol. The relevant data is given in Tab. 3 and Tab. 6. A subset of the randomly sampled machines is shown in Fig. 6b.

**Symbol Interventions.** In the final intervention category, we only consider global explanatory variables. The intervention itself is also global, as the transitions for a given symbol are spread out over the machine we intervene on. For Transformers, the expected length is the most important factor for predicting a high intercept. While for the RNNs the overall machine entropy is the leading explanation. Intuitively, we observe that the more frequent a symbol is in the language the more

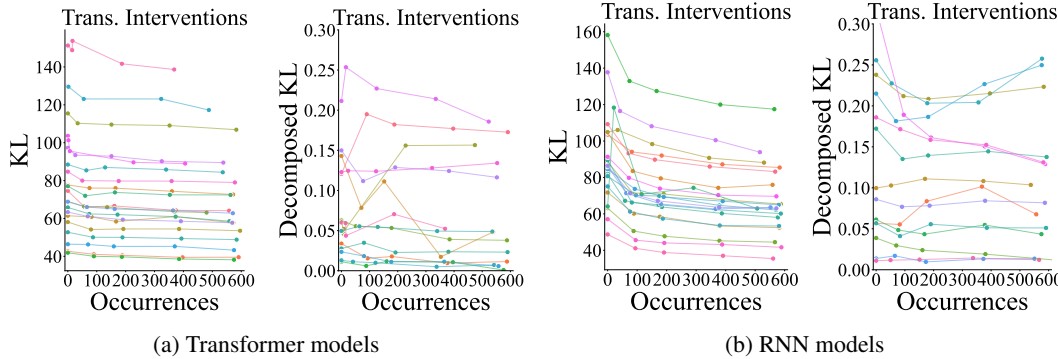

(a) Transformer models                         (b) RNN models

Figure 3: A subset of transition intervention trends, randomly sampled. Each line corresponds to one machine under different intervention constraints.

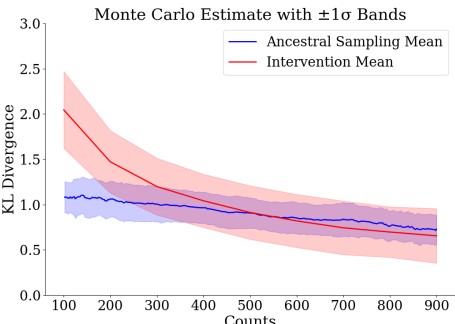

Figure 4: Comparison of decomposed KL under symbol intervention and ancestral sampling.

harmful the intervention is. Much like before, we see clear benefits of increasing the number of symbol occurrences, with the RNNs showing an even stronger added benefit than with other types of interventions. Furthermore, for the RNNs, the machine entropy is a significant predictor of the slope effect, but not so for the Transformer. The relevant data is given in Tab. 4 and Tab. 7. A random subset of the randomly sampled machines is shown in Fig. 7b.

**Decomposed KL.** Although its slope intercept is significant and negative, as observed in a random subset of the randomly sampled machines shown in Fig. 3b, the slope of the decomposed KL is less sensitive to the explanatory variables - global or local. The full KL measures the benefit of increased occurrences on all parts of the modeled language, while the decomposed KL measures only the local effect. We sometimes see a difference in the explanatory effect for the full KL and the decomposed KL. For instance, for state interventions, the forward entropy has the opposite effect. In general, we find that the Transformer models are more sensitive to the global variables and the RNNs to the local variables. We hypothesize that this is due to the Transformer modeling the language more globally at any given timestep, while the RNNs are more concerned about what follows more immediately.

## 7 CAUSAL EFFECT OF SYMBOL OCCURRENCES ON LEARNABILITY

We now conduct a causal study to demonstrate how sampling under intervention can lead to different results than doing ancestral sampling and binning the results afterward. We do this by targeting the property of how often a given symbol occurs. The machines we sample are as before, except we now use 50 states and we increase the probability of a state's chance of accepting to 0.2. For both our causal sampling and ancestral sampling we randomly sample 400 machines each. We then sample 500 strings from each machine and plot the decomposed KL-divergence for the symbol against how often it occurred in the corpus. We evaluate the KL-divergence over 10000 strings to get a good

Monte Carlo estimate of the expected KL-divergence for the causal intervention. See App. H for a derivation of the estimate.

The results are shown in Fig. 4. We see how the Monte Carlo estimate of the estimated decomposed KL-divergence for the symbols when averaged over all of the machines and corpora follows an exponentially declining trend. At the same time, the trend from the ancestral sampling is linear. This clear difference in trends shows exactly why a causal analysis is needed—without it, we would have overestimated the effect of training on a few occurrences and underestimated the effect of including more occurrences.

## 8    RELATED WORK

Several studies have used formal automata as a lens to study neural models (Cleeremans et al., 1989; Jacobsson, 2005; Valvoda et al., 2022; Svete et al., 2024; Borenstein et al., 2024). Theoretical work investigates the representational capacity of neural language models (Merrill, 2023; Strobl et al., 2023). This line of inquiry is part of a broader effort to understand the representational power of neural architectures, such as Transformers (Merrill, 2019; Merrill et al., 2020; Liu et al., 2023). While these studies offer valuable insights into the internal mechanisms of different architectures, the assumptions required for theoretical analysis are often unrealistic and typically provide only an upper or lower bound of what can be practically achieved. This is why the theoretical work is complemented by empirically driven research.

A key component of empirical studies in this field is the use of synthetic datasets. In straightforward cases, these datasets are crafted to investigate specific linguistic phenomena. For instance, the SCAN language and its subsequent adaptations were designed to examine the compositional generalization capabilities of neural models (Lake & Baroni, 2018; Bastings et al., 2018; Ruis et al., 2020). Similarly, k-Dyke languages have been extensively employed to explore the ability of LSTMs to process nested structures(Weiss et al., 2018; Suzgun et al., 2019; Bhattamishra et al., 2020; Hewitt et al., 2020). More recently, Delétang et al. (2023) studied several toy languages to assess inductive biases of neural models in terms of Chomsky hierarchy. By investigating many datasets, Delétang et al. can draw broader conclusions, advancing beyond the single-dataset approaches used in SCAN and k-Dyck language research. A further extension involves studying entire classes of languages, rather than individual datasets (Valvoda et al., 2022; Borenstein et al., 2024). This method has a rich history in grammatical inference studies (Jacobsson, 2005), and lends itself particularly well to linguistic explorations. Our work fits within this broader empirical tradition.

Despite the extensive work on both the theoretical and empirical aspects mentioned above, there has been relatively little focus on using a causal approach to study language model behavior. This gap exists for good reason: causal investigation with natural language is exceptionally challenging, requiring complex taxonomies (Chen et al., 2024) or specific neuron interventions (Vig et al., 2020; Finlayson et al., 2021). We contribute to this work by developing a method to study formal language learning causally.

## 9    CONCLUSION

We have proposed a new methodology for controlled sampling of probabilistic finite state automata, enabling causal probing of the learnability of neural language models. To do so, we introduce the marginal semiring, along with sampling procedures for formal automata that keep track of the number of occurrences that some feature appears—as long as it can be described in terms of groups of transitions that should collectively be targeted for intervention. We demonstrate the applicability of the method with a brief empirical study comparing what local automata properties can predict Transformers and LSTMs learnability. We find that it is not always the same properties of the languages that can predict the learnability of the two architectures—highlighting that there are differences in how they perform sequence modeling. We then show in a causal setting that we get different results when estimating the impact of symbol frequency on symbol learnability if we sample causally or by binning post hoc. Highlighting the importance of using causal methods if one wants to draw causal conclusions.

## REPRODUCIBILITY STATEMENT

The main contributions of this work include an algebraic definition defined in §3, and sampling algorithms defined in §4. Supporting definitions and derivations are given in the appendix, including a closed-form algorithm for the star-operator App. E and derivations showing that the marginal semiring is well formed App. D. The experiments we run are described in detail in §5 and §6, although we note these require non-insignificant GPU resources to reproduce, our experiments took several days to run with NVIDIA H100-equipped GPU nodes. We will publicly release our implementation when appropriate.

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

## A PRELIMINARIES

We now introduce the formal background needed for defining the marginal semiring and causally sampling from it.

### A.1 LANGUAGE MODELS

An **alphabet** $\Sigma$ is a finite non-empty set of **symbols**. The **Kleene closure** $\Sigma^*$ is the set of all strings of symbols from $\Sigma$. A **language model** (LM) $p$ is a probability distribution over $\Sigma^*$. Neural LMs define $p(\boldsymbol{y})$ as a product of next-symbol probability distributions:

$$p(\boldsymbol{y}) \stackrel{\text{def}}{=} p(\text{EOS} \mid \boldsymbol{y}) \prod_{t=1}^{|\boldsymbol{y}|} p(y_t \mid \boldsymbol{y}_{<t}), \tag{8}$$

where $\text{EOS} \notin \Sigma$ is a special end-of-sequence-symbol. We denote $\overline{\Sigma} \stackrel{\text{def}}{=} \Sigma \cup \{\text{EOS}\}$ and $\overline{y}$ an element of $\overline{\Sigma}$. Transformers (Vaswani et al., 2017) and LSTM recurrent neural networks (RNNs) (Elman, 1990; Hochreiter & Schmidhuber, 1997) are popular ways of implementing neural LMs.

### A.2 SEMIRINGS AND WEIGHTED FINITE-STATE AUTOMATA

**Monoid and semiring** We start by introducing some core algebraic concepts.

**Definition A.1** (Monoid). *Let $\mathbb{K}$ be a set, $\odot$ a binary operation, and $\mathbf{1} \in \mathbb{K}$. We say $(\mathbb{K}, \odot, \mathbf{1})$ is a* **monoid** *if (i) $\mathbb{K}$ is closed under $\odot$, (ii) $\odot$ is associative, and (iii) $\mathbf{1}$ is the unit of $\odot$. We say that a monoid is* **commutative** *if $\forall a, b \in \mathbb{K}: \ a \odot b = b \odot a$.*

**Weighted Finite-state Automata.** A **weighted finite-state automaton** (WFSA) $\mathcal{A}$ over a semiring $(\mathbb{K}, \oplus, \otimes, \mathbf{0}, \mathbf{1})$ is a 5-tuple $(\Sigma, Q, \delta, \lambda, \rho)$ where $\Sigma$ is an alphabet, $Q$ is a finite set of states, $\delta$ is a set of weighted transitions rendered as $p \xrightarrow{a/\text{w}} q$ with $p, q \in Q$, $a \in \Sigma$, and $\text{w} \in \mathbb{K}$, and $\lambda\colon Q \to \mathbb{K}$ and $\rho\colon Q \to \mathbb{K}$ are the initial and final weight function, respectively. A **path** $\boldsymbol{\pi}$ in $\mathcal{A}$ is a finite sequence of contiguous transitions, denoted as $q_0 \xrightarrow{a_1/w_1} q_1, \cdots, q_{N-1} \xrightarrow{a_N/w_N} q_N$. The **weight** of $\boldsymbol{\pi}$ is $\boldsymbol{w}(\boldsymbol{\pi}) = w_1 \otimes \cdots \otimes w_N$ and its **yield** is $\sigma(\boldsymbol{\pi}) = a_1 \cdots a_N$. With $\Pi(\mathcal{A})$, we denote the set of all paths in $\mathcal{A}$, and with $\Pi(\mathcal{A}; \boldsymbol{y})$ the subset of all paths in $\mathcal{A}$ with yield $\boldsymbol{y}$. We say that a WFSA $\mathcal{A} = (\Sigma, Q, \delta, \lambda, \rho)$ is **deterministic** (a WDFSA) if, for every $p \in Q, y \in \Sigma$, there is at most one $q \in Q$ such that $p \xrightarrow{a/\text{w}} q \in \delta$ with $\text{w} > 0$, and there is a single state $q_\iota$ with $\lambda(q_\iota) \neq 0$. In such case, we refer to $q_\iota$ as the **initial state**. Naturally, a WDFSA can have at most one path yielding a string $\boldsymbol{y} \in \Sigma^*$ from the initial state $q_\iota$.

## B SEMIRINGS

In §3, we define an abstraction that enables causal interventions on the number of symbols in the dataset produced by a PDFA. In this section, we provide the underlying formalization behind this abstraction. We begin by introducing a basic algebraic structure and a building block of a semiring—the *monoid*.

**Definition B.1.** *(Monoid) Let $\mathbb{K}$ be a set, $\odot$ a binary operation on the set, and $\mathbf{1} \in \mathbb{K}$ be an identity element. We say the the tuple $(\mathbb{K}, \odot, \mathbf{1})$ is a **monoid** if the following properties hold:*

    *(i) $\odot$ is associative: $\forall a, b, c \in \mathbb{K} : (a \odot b) \odot c = a \odot (b \odot c)$;*

    *(ii) $\mathbf{1}$ is the left and right unit: $\forall a \in \mathbb{K} : \mathbf{1} \odot a = a \odot \mathbf{1} = a$;*

    *(iii) $\mathbb{K}$ is closed under $\odot$: $\forall a, b \in \mathbb{K} : a \odot b \in \mathbb{K}$.*

*We say that a monoid is **commutative** if $\forall a, b \in \mathbb{K}: \; a \odot b = b \odot a$.*

We now define a semiring in terms of monoids.

**Definition B.2.** *(Semiring) A **semiring** is a quintuple $(\mathbb{K}, \oplus, \otimes, \mathbf{0}, \mathbf{1})$, where $\mathbb{K}$ is a set equipped with two binary operations $\oplus$ and $\otimes$, such that for all $a, b, c$ in $\mathbb{K}$:*

    *(i) $(\mathbb{K}, \oplus, \mathbf{0})$ is a commutative monoid with identity element $\mathbf{0}$, i.e.,*

- $(a \oplus b) \oplus c = a \oplus (b \oplus c)$
- $\mathbf{0} \oplus a = a \oplus \mathbf{0} = a$
- $a \oplus b = b \oplus a$

    *(ii) $(\mathbb{K}, \otimes, \mathbf{1})$ is a monoid with identity element $\mathbf{1}$, i.e.,*

- $(a \otimes b) \otimes c = a \otimes (b \otimes c)$
- $\mathbf{1} \otimes a = a \otimes \mathbf{1} = a$

    *(iii) Multiplication left and right distributes over addition:*

- $a \otimes (b \oplus c) = (a \otimes b) \oplus (a \otimes c)$
- $(b \oplus c) \otimes a = (b \otimes a) \oplus (c \otimes a)$

    *(iv) Multiplication with $\mathbf{0}$ annihilates $\mathbb{K}$, i.e.,*

- $\mathbf{0} \otimes a = a \otimes \mathbf{0} = \mathbf{0}$

We now introduce the *closed semiring*, a minimal addition that allows us to account for the infinite ways of traversing cyclic automata.

**Definition B.3.** *(Closed semiring) We say that a semiring is a **closed semiring** if there is an additional unary operator * such that for all $a \in \mathbb{K}$*

- $a^* = 1 \oplus a \cdot a^* = 1 \oplus a^* \otimes a$.

Note that if the infinite sum $\bigoplus_{n=0}^{\infty} a^{\otimes n}$ is well-defined and lives in the set $\mathbb{K}$, then it satisfies the two closure axioms given above.

## C    FORMAL PRESENTATION OF FINITE AUTOMATA

We are interested in modeling probabilistic language models (PLMs). We formalize these as weighted finite state automata (WFSA), where the weights correspond to the contextual probabilities of the symbols in the language we sample from the WFSA.

**Definition C.1** (Weighted Finite-State Automaton). *A **weighted finite-state automaton** $\mathcal{A}$ over a semiring $\mathcal{W} = (\mathbb{K}, \oplus, \otimes, \mathbf{0}, \mathbf{1})$ is a 5-tuple $(\Sigma, Q, \delta, \lambda, \rho)$ where*

- *$\Sigma$ is a finite alphabet;*

- *$Q$ is a finite set of states;*

- *$\delta \subseteq Q \times (\Sigma \cup \{\varepsilon\}) \times \mathbb{K} \times Q$ is a finite multi-set of transitions;*

- *$\lambda : Q \to \mathbb{K}$ a weighting function assigning states their initial values;*

- $\rho\colon Q \to \mathbb{K}$ *a weighting function assigning states their final values.*

**Definition C.2.** *(Cyclical Weighted Finite-State Automaton) We say that a weighted finite-state automaton $\mathcal{A}$ is **cyclical** if there exists a sequence of transitions $\delta_1^n, n \in \mathbb{N}$, such that $q_n' = q_0$, and $q_i' = q_{i+1}, \forall i < n$.*

**Definition C.3** (Probabilistic Finite-State Automaton, PFSA). *We say that a WFSA is **probabilistic** if, for all states $q \in Q$, $\delta, \lambda$ and $\rho$ satisfy $\sum_{q \in Q} \lambda(q) = 1$, and $\displaystyle\sum_{q \xrightarrow{y/w} q' \in \delta} w + \rho(q) = 1$.*

**Definition C.4** (Deterministic Probabilistic Finite-State Automaton). *A PFSA $\mathcal{A} = (\Sigma, Q, \delta, \lambda, \rho)$ is **deterministic** if $|\{q \mid \lambda(q) > 0\}| = 1$ and, for every $q \in Q, y \in \Sigma$, there is at most one $q' \in Q$ such that $q \xrightarrow{y/w} q' \in \delta$ with $w > 0$.*

For example, given $\Sigma = \{a_1, a_2\}$, $Q = \{q_1, q_2\}$, and $\mathbb{K} = \{w_1, w_2\}$, we can define a simple cyclical PDFA in Fig. 5, with $\oplus$ and $\otimes$ defined as addition and multiplication over the real numbers.

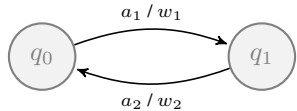

Figure 5: A simple PDFA.

## C.1 LEHMANN'S ALGORITHM

In §5 we sample strings from a WFSA. To do this efficiently, we rely on the backward weights. In general, **backward weights** (also known as backward probabilities or backward values) are used to compute the total weight (e.g. the probability) of paths from a given state to a final state.

**Definition C.5** (Path). *We say that $\boldsymbol{\pi} \subseteq \delta$ is a **path** between $q_1$ and $q_N$ if*

$$\boldsymbol{\pi} = q_1 \xrightarrow{a_1/w_1} q_2, q_2 \xrightarrow{a_2/w_2} q_3, \cdots, q_{N-1} \xrightarrow{a_{N-1}/w_{N-1}} q_N.$$

**Definition C.6** (Path Weight). *The **inner path weight** $w(\boldsymbol{\pi})$ of a path $\boldsymbol{\pi}$ is defined as*

$$\mathrm{w}_{\mathrm{I}}(\boldsymbol{\pi}) \stackrel{\text{def}}{=} \bigotimes_{n=1}^{N-1} w_n. \tag{9}$$

*In the edge case $|\boldsymbol{\pi}| = 0$, we define the inner path weight to be $\mathrm{w}_{\mathrm{I}}(\boldsymbol{\pi}) \stackrel{\text{def}}{=} \mathbf{1}$.*

We are now in a position to define a **backward weight**.

**Definition C.7** (Backward Weight). *Let $\boldsymbol{\beta}_{\mathcal{A}}(q)$ be the sum of the weights of all path weights from $q$ to any final state.*

$$\boldsymbol{\beta}_{\mathcal{A}}(q) \stackrel{\text{def}}{=} \bigoplus_{\substack{\boldsymbol{\pi} \in \Pi(\mathcal{A}), \\ p(\boldsymbol{\pi}) = q}} \mathrm{w}_{\mathrm{I}}(\boldsymbol{\pi}) \otimes \rho(n(\boldsymbol{\pi})) \tag{10}$$

*Where $p(\boldsymbol{\pi})$ and $n(\boldsymbol{\pi})$ denote the origin and the destination states of path $\boldsymbol{\pi}$, respectively. We use $\rho(q)$ for the termination weight at state $q$.*

*When the weights represent probabilities, then $\boldsymbol{\beta}_{\mathcal{A}}(q)$ represents the probability of reaching a final state starting from $q$.*

We use these weights for sampling under interventions in §4. To do so efficiently, and in particularly for cyclical WFSA's, we rely on Lehmann (1977), who defines an algorithm Alg. 1 to efficiently compute the $\oplus$-sum over the paths between any two nodes in a graph, i.e.,

$$\mathbf{R}_{ik} \stackrel{\text{def}}{=} \bigoplus_{\boldsymbol{\pi} \in \Pi(\mathcal{A})(i,k)} \mathrm{w}_{\mathrm{I}}(\boldsymbol{\pi}) \tag{11}$$

In particular, this allows us to use Lehmann's algorithm to compute backward weights using

$$\boldsymbol{\beta}_{\mathcal{A}}(q) = \bigoplus_{\substack{i,k \in Q, \\ p(\boldsymbol{\pi}) = q}} \mathbf{R}_{ik} \otimes \rho(n(\boldsymbol{\pi})) \tag{12}$$

---

**Algorithm 1** Lehmann's algorithm

---

1. **def** Lehmann($\mathbf{M}$):
2.    ▷ $\mathbf{M}$ *is a $D \times D$ matrix over a closed semiring*
3.    $\mathbf{R}^{(0)} \leftarrow \mathbf{M}$
4.    **for** $j \leftarrow 1$ **up to** $D$ :
5.       **for** $i \leftarrow 1$ **up to** $D$ :
6.          **for** $k \leftarrow 1$ **up to** $D$ :
7.             $\mathbf{R}_{ik}^{(j)} \leftarrow \mathbf{R}_{ik}^{(j-1)} \oplus \mathbf{R}_{ij}^{(j-1)} \otimes \left( \mathbf{R}_{jj}^{(j-1)} \right)^* \otimes \mathbf{R}_{jk}^{(j-1)}$
8.    **return** $\mathbf{I} \oplus \mathbf{R}^{(D)}$

---

## D  MARGINAL SEMIRING IS WELL FORMED

Here we provide a derivation to show that the semiring introduced in §3 is well formed, meaning that it satisfies the axioms laid out in App. B. This is also clear from he isomorphism with the truncated polynomial semiring as shown in App. G.

**Proposition D.1.** *(Marginal semirings are well formed) The marginal semiring (Def. 3.2) is well formed.*

*Proof.* We need to show that the semiring axioms hold. Let $\mathbf{v}, \mathbf{v}', \mathbf{v}'' \in \mathbb{K}^{N+1}$

(i)  $(\mathbb{K}^{N+1}, \oplus)$ is a commutative monoid:

- $\oplus$ is associative:

$$(\mathbf{v} \oplus \mathbf{v}') \oplus \mathbf{v}''_i = (\mathbf{v}_i + \mathbf{v}'_i) + \mathbf{v}''_i \tag{13}$$
$$= \mathbf{v}_i + (\mathbf{v}'_i + \mathbf{v}''_i) \qquad + \text{ is associative} \tag{14}$$
$$= \mathbf{v}_i \oplus (\mathbf{v}' \oplus \mathbf{v}'')_i \tag{15}$$

- $\oplus$ is commutative:

$$(\mathbf{v} \oplus \mathbf{v}')_i = \mathbf{v}_i + \mathbf{v}'_i = \mathbf{v}'_i + \mathbf{v}_i = (\mathbf{v} \oplus \mathbf{v}')_i \tag{16}$$

- $\mathbf{0}$ is a left and right unit:

$$(\mathbf{0} \oplus \mathbf{v})_i = \mathbf{0}_i + \mathbf{v}_i = 0 + \mathbf{v}_i = \mathbf{v}_i \tag{17}$$
$$(\mathbf{v} \oplus \mathbf{0})_i = \mathbf{v}_i + \mathbf{0}_i = \mathbf{v}_i + 0 = \mathbf{v}_i \tag{18}$$

- $\mathbb{K}^{N+1}$ is closed under $\oplus$:

$$(\mathbf{v} + \mathbf{v}')_i = \mathbf{v}_i + \mathbf{v}'_i \in \mathbb{K} \implies (\mathbf{v} + \mathbf{v}') \in \mathbb{K}^{N+1} \tag{19}$$

(ii)  $(\mathbb{K}^{N+1}, \otimes)$ is a monoid: Since $(\mathbb{K}, \times)$ is a monoid, we have that

- $\otimes$ is associative:

$$((\mathbf{v} \otimes \mathbf{v}') \otimes \mathbf{v}'')_i = \sum_{m=0}^{i} (\mathbf{v} \otimes \mathbf{v}')_m \times \mathbf{v}''_{i-m} \tag{20}$$

$$= \sum_{m=0}^{i} \left( \sum_{n=0}^{m} \mathbf{v}_n \times \mathbf{v}'_{m-n} \right) \times \mathbf{v}''_{i-m} \tag{21}$$

$$= \sum_{m=0}^{i} \sum_{n=0}^{m} (\mathbf{v}_n \times \mathbf{v}'_{m-n}) \times \mathbf{v}''_{i-m} \qquad \times \text{ is distributive over } + \tag{22}$$

$$= \sum_{m=0}^{i} \sum_{n=0}^{m} \mathbf{v}_n \times (\mathbf{v}'_{m-n} \times \mathbf{v}''_{i-m}) \qquad \times \text{ is associative} \tag{23}$$

$$= \sum_{n=0}^{i} \mathbf{v}_n \times \sum_{m=n}^{i} (\mathbf{v}'_{m-n} \times \mathbf{v}''_{i-m}) \qquad \times \text{ is distributive over } + \tag{24}$$

$$= \sum_{n=0}^{i} \mathbf{v}_n \times \sum_{m'=0}^{i-n} (\mathbf{v}'_{m'} \times \mathbf{v}''_{i-n-m'}) \qquad m' = m - n \tag{25}$$

$$= \sum_{n=0}^{i} \mathbf{v}_n \times ((\mathbf{v}' \otimes \mathbf{v}'')_{i-n}) \qquad \text{definition of } \otimes \tag{26}$$

$$= (\mathbf{v} \otimes (\mathbf{v}' \otimes \mathbf{v}''))_i \qquad \text{definition of } \otimes \tag{27}$$

$$\tag{28}$$

- $\mathbf{1}$ is a left and right unit, by Def. 3.2 *(iv)* we have:

$$(\mathbf{1} \otimes \mathbf{v})_j = \sum_{n=0}^{j} \mathbf{1}_n \times \mathbf{v}_{j-n} = 1 \times \mathbf{v}_j = \mathbf{v}_j \tag{29}$$

$$(\mathbf{v} \otimes \mathbf{1})_j = \sum_{n=0}^{j} \mathbf{v}_n \times 1_{j-n} = \mathbf{v}_j \times 1 = \mathbf{v}_j \tag{30}$$

- $\mathbb{K}^{N+1}$ is closed under $\otimes$:

$$(\mathbf{v} \otimes \mathbf{v}')_j = \sum_{n=0}^{j} \mathbf{v}_n \times \mathbf{v}'_{j-n} \in \mathbb{K} \implies (\mathbf{v} \otimes \mathbf{v}') \in \mathbb{K}^{N+1} \tag{31}$$

(iii) Multiplication with $\mathbf{0}$ annihilates $\mathbb{K}^{N+1}$:

$$(\mathbf{0} \otimes \mathbf{v})_j = \sum_{n=0}^{j} \mathbf{0}_n \times \mathbf{v}_{j-n} = \sum_{n=0}^{j} 0 \times \mathbf{v}_{j-n} = 0 = \mathbf{0}_j \tag{32}$$

$$(\mathbf{v} \otimes \mathbf{0})_j = \sum_{n=0}^{j} \mathbf{v}_n \times \mathbf{0}_{j-n} = \sum_{n=0}^{j} \mathbf{v}_n \times 0 = 0 = \mathbf{0}_j \tag{33}$$

(iv) Multiplication left and right distributes over addition.

- From the left:

$$(\mathbf{v} \otimes (\mathbf{v}' \oplus \mathbf{v}''))_i = \sum_{n=0}^{i} \mathbf{v}_n \times (\mathbf{v}' \oplus \mathbf{v}'')_{i-n} \tag{34}$$

$$= \sum_{n=0}^{i} \mathbf{v}_n \times (\mathbf{v}'_{i-n} + \mathbf{v}''_{i-n}) \tag{35}$$

$$= \sum_{n=0}^{i} \mathbf{v}_n \times \mathbf{v}'_{i-n} + \mathbf{v}_n \times \mathbf{v}''_{i-n} \qquad \text{distributivity of base semiring} \tag{36}$$

$$= \sum_{n=0}^{i} \mathbf{v}_n \times \mathbf{v}'_{i-n} + \sum_{n=0}^{i} \mathbf{v}_n \times \mathbf{v}''_{i-n} \tag{37}$$

$$= (\mathbf{v} \otimes \mathbf{v}')_i + (\mathbf{v} \otimes \mathbf{v}'')_i \tag{38}$$

$$= ((\mathbf{v} \otimes \mathbf{v}') \oplus (\mathbf{v} \otimes \mathbf{v}''))_i \tag{39}$$

- The other direction can be derived with minimal modifications.

(v) The marginal semiring over $\mathbb{K}^{N+1}$ is closed under the *-operator. We have from Prop. E.1 below, including the definition of $C$ in terms of $\mathbf{v}$:

$$(\mathbf{v}^*)_i = \mathbf{v}_0^{\star} \times \left( \mathbf{1}_i + \sum_{n=1}^{i} \mathbf{v}_n \times \mathbf{v}^*{}_{i-n} \right) = \mathbf{v}_0^{\star} \times C \in \mathbb{K} \implies \mathbf{v}^* \in \mathbb{K}^{N+1} \tag{40}$$

$\blacksquare$

# E  Closed form solution

**Lemma E.1.** *(Arden's rule for semirings)  Given a semiring $(\mathbb{K}, \oplus, \otimes, \mathbf{0}, \mathbf{1})$, and $X, A, B$ in $\mathbb{K}$, it holds that*

$$X = AX \oplus B \implies X = A^*B \tag{41}$$

*This result is commonly known as Arden's rule[2]. In its more common form, it states that the above holds for regular languages. Here, we show that it holds more generally in the context of a semiring.*

*Proof.*

$$X = AX \oplus B \implies X = A(AX \oplus B) \oplus B \tag{42}$$

$$\implies X = A(A(AX \oplus B) \oplus B) \oplus B \tag{43}$$

$$\implies X = \left( \bigoplus_{i=0}^{n} A^i \right) B \oplus A^n X \qquad \text{insert for } X, n\text{-times} \tag{44}$$

$$n \to \infty \implies X = \left( \bigoplus_{i=0}^{\infty} A^i \right) B \qquad (*) \tag{45}$$

$$\implies X = A^*B \qquad \text{def. of } A^* \tag{46}$$

Where the second term disappears in $(*)$ when we take the limit. This derivation is a reformulation of that given by Miles (2016). Importantly, we also need to make sure that the limit is well defined in $(*)$, meaning that the solution is minimal in the sense that any other solution to the equation contains $A^*B$ in it.

---

[2]See https://en.wikipedia.org/wiki/Arden%27s_rule.

Let's assume that $A^*B$ is not a minimal solution, meaning that $Y = A^nB$ for some $n \in \mathbb{N}_+$ is a solution. We also have, per the first part of the derivation above, that $Y$ must be of the form

$$Y = \left( \bigoplus_{i=0}^{n} A^i \right) B \oplus A^n Y \tag{47}$$

$$= \left( \bigoplus_{i=0}^{n} A^i \right) B \oplus A^n (A^n B) \tag{48}$$

$$= \left( \bigoplus_{i=0}^{n} A^i \right) B \oplus A^{2n} B \tag{49}$$

The last term in the last equation of the derivation contradicts the assumption that $Y = A^n B$ is a minimal solution, showing that the minimal solution must be of the form $A^*B$.

∎

**Proposition E.1.** *As the unary \*-operator of the marginal semiring is defined in terms of an infinite sum, a closed-form calculation of it is desired. For any marginal semiring over $\mathbb{K}^{N+1}$, with $\mathbf{v} \in \mathbb{K}^{N+1}$, we state that, for all $1 < i \leqslant N+1$:*

$$\mathbf{v}^*{}_i = \mathbf{v}_0^\star \times \left( \mathbf{1}_i + \sum_{n=1}^{i} \mathbf{v}_n \times \mathbf{v}^*{}_{i-n} \right) \tag{50}$$

*Proof.* We have

$$(\mathbf{v}^*)_i = \left( \bigoplus_{n=0}^{\infty} \mathbf{v}^{\otimes n} \right)_i \tag{51a}$$

$$= \mathbf{1}_i + \left( \mathbf{v} \otimes \bigoplus_{n=0}^{\infty} \mathbf{v}^{\otimes n} \right)_i \tag{51b}$$

$$= \mathbf{1}_i + (\mathbf{v} \otimes \mathbf{v}^*)_i \tag{51c}$$

$$= \mathbf{1}_i + \sum_{n=0}^{i} \mathbf{v}_n \times (\mathbf{v}^*)_{i-n} \tag{51d}$$

$$= \mathbf{1}_i + \mathbf{v}_0 \times (\mathbf{v}^*)_i + \sum_{n=1}^{i} \mathbf{v}_n \times (\mathbf{v}^*)_{i-n} \tag{51e}$$

$$= \mathbf{v}_0 \times (\mathbf{v}^*)_i + \underbrace{\mathbf{1}_i + \sum_{n=1}^{i} \mathbf{v}_n \times (\mathbf{v}^*)_{i-n}}_{\overset{\text{def}}{=} C} \tag{51f}$$

$$= \mathbf{v}_0 \times (\mathbf{v}^*)_i + C \tag{51g}$$

Making use of the fact that $C$ relies only on $\mathbf{v}^*{}_j$, for $j < i$, Lemma E.1, we get that the above equation has the following solution:

$$(\mathbf{v}^*)_i = (\mathbf{v}_0)^* \times C \tag{51h}$$

$$= (\mathbf{v}_0)^* \times \left( \mathbf{1}_i + \sum_{n=1}^{i} \mathbf{v}_n \times (\mathbf{v}^*)_{i-n} \right), \tag{51i}$$

which is what we wanted to show. ∎

Prop. E.1 gives us a closed formulation for the star value of any element in a marginal semiring. A straightforward implementation of it gives us $\mathcal{O}(i^3)$ runtime; the convolution gives us a linear factor and the $\mathbf{v}^*$ the squared factor. We can further speed this up with memoization by storing the intermediate calculations, giving us $\mathcal{O}(i^2)$ runtime. Below, we show how this can be even further improved.

### E.1 Speed-ups when Lifting the Real Semiring

The multiplication operation in the marginal semiring is a signification bottleneck in the applications we consider. A common trick for speeding up convolutions is the fast Fourier transform (FFT). Put succinctly, the FFT turns convolutions in the original domain into pointwise multiplication in the target domain. The former is commonly referred to as the time domain and the latter as the frequency domain. By using the FFT we can thus calculate $\mathbf{v}^*$ in $\mathcal{O}(N \log N)$-time for $\mathbf{v} \in \mathbb{K}^{N+1}$.

### E.2 Runtime of occurrence sampling

We provide additional details on the runtime of Thm. 4.2 below.

(1) Occurrence sampling: We first sample the number of occurrences of the property we target, i.e. how often the property should occur in each of the $K$ strings. We need to do $K$ convolutions each with the cost of a convolution, $n\log(n)$, giving us $K \cdot n\log(n)$. Even if we store the prior result for practical gains this does not improve the big-O.

(2) Property occurrence sampling: We first need to calculate the pathsum (pathsum) of all states. The pathsum computation using Lehmann's algorithm requires $O(|Q|^3)$ operations (since we need to do $|Q|$ iterations in the calculations for $|Q|$ states and max $|Q|$ transitions in a fully connected graph). Then for each of these operations, we need to do the multiplication over a vector of size $n + 1$, which we can do in $n\log(n)$ using the FFT approach. So we get $O(|Q|^3 n\log(n))$. Then we sample a symbol for each step in the max length, let's call this $L$, and we have $K$ strings for which we need to do a convolution each so we get $O(KLn\log(n))$. This means the pathsum calculations dominate unless $KL > |Q|^3$.

## F    Proofs

**Theorem 3.1** (Path Weight Interpretation). *Let $\phi$ be a feature function and $\mathcal{A}_{\mathcal{L}_\phi}$ its marginal automaton. We denote the number of times a feature occurs on a path $\boldsymbol{\pi}$ as $|\boldsymbol{\pi}|_\phi$. If $\boldsymbol{\pi}$ is a path in $\mathcal{A}$, and $\mathrm{w}_\mathrm{I}(\boldsymbol{\pi})$ is the path weight, the following holds:*

$$|\boldsymbol{\pi}|_\phi = \operatorname*{argmax}_{1 \leqslant i \leqslant N} \mathrm{w}_\mathrm{I}(\boldsymbol{\pi})_i \ \text{ and } \forall j \neq |\boldsymbol{\pi}|_\phi, \mathrm{w}_\mathrm{I}(\boldsymbol{\pi})_j = 0 \tag{1}$$

*In words, the index of the only non-zero element of $\mathrm{w}_\mathrm{I}(\boldsymbol{\pi})$ tells us how often the feature occurs in $\boldsymbol{\pi}$.*

*Proof.* We proceed by induction over the length of the path. If the path has a single element, then the path weight is the lifted weight, and the result follows directly. Let us now assume that the hypothesis holds for a path $\boldsymbol{\pi}'$ of length $n$, i.e., the target feature occurs $i$ times and $\mathrm{w}_\mathrm{I}(\boldsymbol{\pi}')_i$ is the only non-zero value in the path-weight. Let $\boldsymbol{\pi}$ be a path of length $n + 1$, and $\boldsymbol{\pi}'$ be the path with the first $n$ elements, we then have

$$\mathrm{w}_\mathrm{I}(\boldsymbol{\pi}) = \bigotimes_{n=1}^{N} w_n \tag{52}$$

$$= \mathrm{w}_\mathrm{I}(\boldsymbol{\pi}') \otimes w_N \qquad \text{by assumption} \tag{53}$$

If $w_N$ does not result in the target feature, then $\operatorname{argmax}_{1 \leqslant i \leqslant N} \mathrm{w}_\mathrm{I}(\boldsymbol{\pi})_i = \operatorname{argmax}_{1 \leqslant i \leqslant N} \mathrm{w}_\mathrm{I}(\boldsymbol{\pi}')_i$ since the only non zero value is $(w_N)_0$. If the feature is observed, then we have

$$\mathrm{w}_\mathrm{I}(\boldsymbol{\pi})_j = (\mathrm{w}_\mathrm{I}(\boldsymbol{\pi}') \otimes w_N)_j \tag{54}$$

$$= \sum_{m=0}^{j} \mathrm{w}_\mathrm{I}(\boldsymbol{\pi}')_m \times (w_N)_{j-m} \qquad \text{by definition} \tag{55}$$

$$= \mathrm{w}_\mathrm{I}(\boldsymbol{\pi}')_i \times (w_N)_{j-i} \qquad \text{by assumption} \tag{56}$$

$$= \begin{cases} |\boldsymbol{\pi}'|_\phi \times (w_N)_1 & j = i + 1 \\ |\boldsymbol{\pi}'|_\phi \times 0 & j \neq i + 1 \end{cases} \tag{57}$$

We have shown that the only non-zero element is the $(i + 1)$-the one, and by assumption, that its position corresponds to how many occurrences of the target feature were seen as part of traversing the path. ∎

**Theorem 3.2** (Pathsum Interpretation). *Let $\mathcal{A}$ be a PFSA and $\Pi$ be a random variable over the paths in $\mathcal{A}_{\mathcal{L}_\phi}$. Then, $|\Pi|_\phi$ is also a random variable and we have*

$$p(|\Pi|_\phi = n) = \boldsymbol{\beta}_{\mathcal{A}_{\mathcal{L}_\phi}}(q)_n. \tag{2}$$

*In words, the probability of exactly $n$ occurrences of the feature in the string scanned by a randomly sampled path is the $n$-th element of the backward weight for $n \in \{0, 1, \ldots, N\}$.*

*Proof.* Assume $q$ is the only start state. We then have

$$p(|\Pi|_\phi = n) = \sum_{|\boldsymbol{\pi}|_\phi = n} p(\boldsymbol{\pi}) \tag{58}$$

$$= \sum |\mathrm{w}_{\mathrm{I}}(\boldsymbol{\pi})|_n \tag{59}$$

$$= \boldsymbol{\beta}_{\mathcal{A}_{\mathcal{L}_\phi}}(q)_n \qquad \text{by definition of } \boldsymbol{\beta}_{\mathcal{A}_{\mathcal{L}_\phi}} \tag{60}$$

We also use $Z_n \overset{\text{def}}{=} \boldsymbol{\beta}_{\mathcal{A}_{\mathcal{L}_\phi}}(q)_n$, when $q$ is the only start state. $\blacksquare$

**Theorem 4.1** (Probability over set of strings). *Let $(\mathrm{K}_i)_{i \in I}$ be a set of indexed strings sampled from a PFSA, and $|(\mathrm{K}_i)_{i \in I}|_\phi$ denote the number of occurrences of the feature in all strings combined. The probability of seeing $n$ occurrences from $\phi$ in $(\mathrm{K}_i)_{i \in I}$ is given by*

$$P(|(\mathrm{K}_i)_{i \in I}|_\phi = n) = (Z^{\otimes k})_n, \tag{3}$$

*where $k = |I|$ and $Z$ is the pathsum of the marginal semiring acquired by lifting the automaton while targeting the features.*

*Proof.* We proceed by induction over the size of the set $(\mathrm{K}_i)_{i \in I}$. If there is a single string, $k = 1$, then the equation holds by Thm. 3.2. Assuming the hypothesis for some $k > 1$, where $\mathrm{K}_\kappa$ is some string in $(\mathrm{K}_i)_{i \in I}$, we have

$$P(|(\mathrm{K}_i)_{i \in I}|_\phi = n) = \sum_{m=0}^{n} P(|(\mathrm{K}_i)_{i \in I \setminus \{\kappa\}}|_\phi = m) \cdot P(|\mathrm{K}_\kappa|_\phi = n - m) \qquad (*) \tag{61}$$

$$= \sum_{m=0}^{n} (Z^{\otimes k-1})_m \cdot Z_{n-m} \qquad (**) \tag{62}$$

$$= (Z^{\otimes k-1} \otimes Z)_n \qquad (***) \tag{63}$$

$$= (Z^{\otimes k})_n. \tag{64}$$

Where we in $(*)$ use that the sampling is independent, in $(**)$ by the induction hypothesis and Thm. 3.2, finally $(***)$ follows by the definition of $\otimes$. $\blacksquare$

**Theorem 4.2** (Sampling lengths). *Let $Z \in \mathbb{R}^{N+1}$ be the pathsum of the lifted marginal automaton $\mathcal{A}_{\mathcal{L}_\phi}$, corresponding to some PFSA we wish to sample from, for some target features $\phi$. Let $\mathrm{K}_k$ be the $k$-th string sampled. Assuming that we have assigned $m$ out of $N$ symbols to the first $k - 1$ sampled strings, then the probability of seeing $n$ symbols in the next string is given by*

$$p(|\mathrm{K}_k|_\phi = n) = Z_n \cdot (Z^{\otimes K-k-1})_{N-m-n}. \tag{4}$$

*Proof.* Since the sampling of strings is independent, we can write

$$P(|\mathrm{K}_k|_\phi = n) = P(|\mathrm{K}_k|_\phi = n) \cdot P(|(\mathrm{K}_{>k})|_\phi = N - m - n) \tag{65}$$

$$= Z_n \cdot P(|(\mathrm{K}_{>k})|_\phi = N - n - m) \qquad \text{Thm. 3.2} \tag{66}$$

$$= Z_n \cdot (Z^{\otimes K-k-1})_{N-m-n} \qquad \text{Thm. 4.1.} \tag{67}$$

Which is what we wanted to show. $\blacksquare$

# G  Isomorphism to the truncated polynomial semiring

Here we show that the marginal semiring of a given order is equivalent to a truncated polynomial semiring.

**Theorem G.1.** *The marginal semiring of order $N$ over $\mathbb{K}$ is isomorphic to the truncated polynomial ring $\mathbb{K}[x]/(x^{N+1})$.*

*Proof.* let $\left(\mathbb{K}^{N+1}, \oplus, \otimes, \mathbf{0}, \mathbf{1}\right)$ be the marginal semiring over base semiring $\mathbb{K}$, and let $\mathbb{K}[x]/(x^{N+1})$ be the corresponding quotient ring of polynomials. We then define $\phi : \mathbb{K}^{N+1} \to \mathbb{K}[x]/(x^{N+1})$ by:

$$\phi([a_0, a_1, \ldots, a_N]) = \sum_{i=0}^{N} a_i x^i \tag{68}$$

We need to show that $\phi$ is a homeomorphic bijection.

The bijectivity is clear since $a_0, a_1, \ldots, a_N$ each corresponds to a coefficient in the polynomial, and these collectively uniquely determine a polynomial and an element in $\mathbb{K}^{K+1}$.

Addition is preserved since

$$\phi(\mathbf{v} \oplus \mathbf{w}) = \phi([v_0 + w_0, \ldots, v_N + w_N]) = \sum_{i=0}^{N} (v_i + w_i)x^i = \phi(\mathbf{v}) + \phi(\mathbf{w}) \tag{69}$$

We now show that multiplication is preserved. Let $\mathbf{v}, \mathbf{v}' \in \mathbb{K}^{N+1}$ be vectors in the marginal semiring. The product of these polynomials in $\mathbb{K}[x]/(x^{N+1})$ is:

$$\phi(\mathbf{v}) \cdot \phi(\mathbf{v}') = \left(\sum_{i=0}^{N} v_i x^i\right) \cdot \left(\sum_{j=0}^{N} v'_j x^j\right) \tag{70}$$

For any $k \leqslant N$, the coefficient of $x^k$ in this product is:

$$\sum_{i+j=k} v_i v'_j = \sum_{i=0}^{k} v_i v'_{k-i} \tag{71}$$

In the counting semiring, the convolution $\mathbf{v} \otimes \mathbf{v}'$ is defined component-wise as:

$$(\mathbf{v} \otimes \mathbf{v}')_k = \sum_{i=0}^{k} v_i v'_{k-i} \tag{72}$$

Therefore, for all $k \leqslant N$:

$$\phi(\mathbf{v} \otimes \mathbf{v}')_k = (\phi(\mathbf{v}) \cdot \phi(\mathbf{v}'))_k \tag{73}$$

And finally, it's clear that $\phi(\mathbf{0}) = 0$ and $\phi(\mathbf{1}) = 1$. ∎

# H  Decomposing the KL divergence by transitions

Let $p_A$ be a probabilistic finite automaton generating sequences over some alphabet. Each sequence $x$ decomposes into transitions, where each transition $\delta$ consists of state $q$, symbol $\sigma_i$, and weight $w$. Given a language model $p_\theta$ and a set of transitions of interest , we decompose the KL divergence to analyze how well $p_\theta$ captures these transitions. At each step, $p_A$ takes transition $\delta$ with probability $w$, while $p_\theta$ predicts the next symbol given the history $\sigma_{(<i)}$ of all symbols preceding position $i$. Theoretically, this decomposition can be written as:

$$D_{\mathrm{KL}}(p_{\mathcal{A}}\|p_{\boldsymbol{\theta}}) = \mathbb{E}_{x \sim p_{\mathcal{A}}}\left[\log \frac{p_{\mathcal{A}}(x)}{p_{\boldsymbol{\theta}}(x)}\right] \tag{74}$$

$$= \mathbb{E}_{x \sim p_{\mathcal{A}}}\left[\log \frac{\prod_{i=1}^{|x|} p_{\mathcal{A}}(\sigma_i \mid \sigma_{(<i)})}{\prod_{i=1}^{|x|} p_{\boldsymbol{\theta}}(\sigma_i \mid \sigma_{(<i)})}\right] \tag{75}$$

$$= \mathbb{E}_{x \sim p_{\mathcal{A}}}\left[\sum_{i=1}^{|x|} \log \frac{p_{\mathcal{A}}(\sigma_i \mid \sigma_{(<i)})}{p_{\boldsymbol{\theta}}(\sigma_i \mid \sigma_{(<i)})}\right] \tag{76}$$

$$= \sum_{\delta \in \cup^c} \mathbb{E}_{\sigma_{(<i)} \sim p_{\mathcal{A}}}\left[\log \frac{p_{\mathcal{A}}(\sigma_i \mid \sigma_{(<i)})}{p_{\boldsymbol{\theta}}(\sigma_i \mid \sigma_{(<i)})}\right] \tag{77}$$

$$= \underbrace{\sum_{\delta \in} \mathbb{E}_{\sigma_{(<i)} \sim p_{\mathcal{A}}}\left[\log \frac{w}{p_{\boldsymbol{\theta}}(\sigma_i \mid \sigma_{(<i)})}\right]}_{\text{Target transitions}} + \underbrace{\sum_{\delta \in^c} \mathbb{E}_{\sigma_{(<i)} \sim p_{\mathcal{A}}}\left[\log \frac{w}{p_{\boldsymbol{\theta}}(\sigma_i \mid \sigma_{(<i)})}\right]}_{\text{Other transitions}} \tag{78}$$

And the empirical version

$$D_{\mathrm{KL}}(p_{\mathcal{A}}\|p_{\boldsymbol{\theta}}) = \sum_{n=1}^{K} p_{\mathcal{A}}(\overline{\sigma}_n) \log \frac{p_{\mathcal{A}}(\overline{\sigma}_n)}{p_{\boldsymbol{\theta}}(\overline{\sigma}_n)} \tag{79}$$

$$= \sum_{n=1}^{K} p_{\mathcal{A}}(\overline{\sigma}_n) \log \frac{\prod_{i=1}^{|\overline{\sigma}_n|} p_{\mathcal{A}}(\sigma_i \mid \sigma_{(<i)})}{\prod_{i=1}^{|\overline{\sigma}_n|} p_{\boldsymbol{\theta}}(\sigma_i \mid \sigma_{(<i)})} \tag{80}$$

$$= \sum_{n=1}^{K} p_{\mathcal{A}}(\overline{\sigma}_n) \sum_{i=1}^{|\overline{\sigma}_n|} \log \frac{p_{\mathcal{A}}(\sigma_i \mid \sigma_{(<i)})}{p_{\boldsymbol{\theta}}(\sigma_i|\sigma_{(<i)})} \tag{81}$$

$$= \sum_{\delta \in \cup^c} \sum_{(n,i) \in \mathcal{O}_{\delta}} p_{\mathcal{A}}(\overline{\sigma}_n) \log \frac{p_{\mathcal{A}}(\sigma_i \mid \sigma_{(<i)})}{p_{\boldsymbol{\theta}}(\sigma_i|\sigma_{(<i)})} \tag{82}$$

$$= \underbrace{\sum_{\delta \in} \sum_{(n,i) \in \mathcal{O}_{\delta}} p_{\mathcal{A}}(\overline{\sigma}_n) \log \frac{w}{p_{\boldsymbol{\theta}}(\sigma_i \mid \sigma_{(<i)})}}_{\text{Target transitions}} \tag{83}$$

$$+ \underbrace{\sum_{\delta \in^c} \sum_{(n,i) \in \mathcal{O}_{\delta}} p_{\mathcal{A}}(\overline{\sigma}_n) \log \frac{w}{p_{\boldsymbol{\theta}}(\sigma_i \mid \sigma_{(<i)})}}_{\text{Other transitions}} \tag{84}$$

Where $\mathcal{O}_{\delta}$ represents all positions where transition $\delta$ appears in our sampled sequences, i.e. $\mathcal{O}_{\delta} \overset{\text{def}}{=} \{(n,i) : \text{transition } \delta \text{ occurs at position } i \text{ in sequence } \overline{\sigma}_n\}$

But if we have already sampled the strings from $\mathcal{A}$ it suffices to calculate

$$D_{\mathrm{KL}}(p_{\mathcal{A}}\|p_{\boldsymbol{\theta}}) = \underbrace{D_{\mathrm{KL}}(p_{\mathcal{A}}\|p_{\boldsymbol{\theta}} \mid)}_{\text{Target transitions}} + \underbrace{D_{\mathrm{KL}}(p_{\mathcal{A}}\|p_{\boldsymbol{\theta}} \mid {}^c)}_{\text{Other transitions}} \tag{85}$$

$$= \underbrace{\sum_{(n,i) \in \mathcal{O}} \log \frac{w}{p_{\boldsymbol{\theta}}(\sigma_i|\sigma_{(<i)})}}_{\text{Target transitions}} + \underbrace{\sum_{(n,i) \in \mathcal{O}_c} \log \frac{w}{p_{\boldsymbol{\theta}}(\sigma_i|\sigma_{(<i)})}}_{\text{Other transitions}} \tag{86}$$

We can then constrain this decomposition to exact transitions relevant to our three interventions. We simply limit the samples we marginalize over: If we target a symbol, we only include the data points

containing the target symbol in the first position. If we target a single transition, we only include the entries corresponding to that transition, i.e., where the symbol, source, and target state are those we are interested in. For state interventions, we only consider the elements where the target state is the intervention state. We report results as the average divergences over the held-out samples.

Where $\mathcal{A}$ is sampled from some distribution $\mathbb{A}$. If we randomly sample these as a Monte Carlo estimate the target we get

$$\mathbb{E}_A[\mathrm{D}_{\mathrm{KL}}(p_\mathcal{A}\|p_\theta)] = \mathbb{E}_A\left[\mathrm{D}_{\mathrm{KL}}(p_\mathcal{A}\|p_\theta \mid) + \mathrm{D}_{\mathrm{KL}}(p_\mathcal{A}\|p_\theta \mid {}^c)\right] \tag{87}$$

$$= \mathbb{E}_A\left[\sum_{(n,i)\in\mathcal{O}}\log\frac{w}{p_\theta(\sigma_i|\sigma_{(<i)})} + \sum_{(n,i)\in\mathcal{O}_c}\log\frac{w}{p_\theta(\sigma_i|\sigma_{(<i)})}\right] \tag{88}$$

$$= \sum_j p(\mathcal{A}_j)\left[\sum_{(n,i)\in\mathcal{O}^j}\log\frac{w_j}{p_\theta(\sigma_i|\sigma_{(<i)})} + \sum_{(n,i)\in\mathcal{O}_c^j}\log\frac{w_j}{p_\theta(\sigma_i|\sigma_{(<i)})}\right] \tag{89}$$

$$\approx \frac{1}{J}\sum_{j=1}^{J}\left[\frac{1}{|\mathcal{O}^j|}\sum_{(n,i)\in\mathcal{O}^j}\log\frac{w_j}{p_\theta(\sigma_i|\sigma_{(<i)})} + \frac{1}{|\mathcal{O}_c^j|}\sum_{(n,i)\in\mathcal{O}_c^j}\log\frac{w_j}{p_\theta(\sigma_i|\sigma_{(<i)})}\right] \tag{90}$$

where $\mathcal{A}_j \sim \mathbb{A}$ and $J$ is the number of sampled machines. The targeted decomposed KL we calculate is thus given by the Monte Carlo estimate

$$\mathrm{D}_{\mathrm{KL}\,\mathrm{targeted}} \approx \frac{1}{J}\sum_{j=1}^{J}\left[\frac{1}{|\mathcal{O}^j|}\sum_{(n,i)\in\mathcal{O}^j}\log\frac{w_j}{p_\theta(\sigma_i|\sigma_{(<i)})}\right] \tag{91}$$

Which also serves as an estimate when we intervene on property $\pi_k$, set to $N$, as in

$$\mathrm{D}_{\mathrm{KL}\,\mathrm{targeted}} \approx \mathbb{E}_{\mathcal{A}\sim\mathbb{A}}\left[\mathbb{E}_{\sigma_k\sim P(\sigma_k|do(\pi_k=N))}\left[\log\frac{w}{p_\theta(\sigma|\sigma_{(<)})}\right]\right]. \tag{92}$$

## I  DIRICHLET SAMPLING

We use Dirichlet sampling to sample the weights of the PDFAs used in our experiments. Its main benefit is that we can readily sample values that add up to 1 and thus provide a probability distribution. The process works as follows: a sample $\mathbf{x} = x_1, \ldots, x_n$ is drawn by first i.i.d. sampling $y_i \sim \mathrm{Gamma}(1, 1)$ for $i = 1, \ldots, k$ using a uniform parameterization. The samples are then given by $x_i = \frac{y_i}{\sum y_j}$, ensuring that $\sum x_i = 1$.

## J  NEURAL LANGUAGE MODELS

We keep a fixed maximum sequence length of 256, a learning rate of 0.001, 6 layers, an embedding size of 64, the number of hidden units in the feedforward layers of the Transformer is 256, the hidden dimension for the RNN is 64, dropout of 0.2 is used, and gradient clipping with a threshold of 0.25. We use 4 attention heads per layer and initialize the range of the weights with a standard deviation of 0.1. For the RNNs, we train for 4 epochs and 10 for the Transformers, logging the best result over the epochs. Each symbol is directly mapped to a corresponding token. Special beginning-of-sentence (BOS) and end-of-sentence (EOS) tokens are also used. We set the batch size to 32 during training and used the Adam optimizer. This exactly follows the configuration used by Borenstein et al. (2024).

Table 1: Adjusted $R^2$ values for the secondary linear models, showing how much of the variance is explained by the explanatory variables. The first value in the pair is for the intersect, and the second for the slope.

|  | Transformer | | RNN | |
|---|---|---|---|---|
|  | KL | Decomp. KL | KL | Decomp. KL |
| Transition | 0.92/0.77 | 0.41/0.45 | 0.93/0.75 | 0.49/0.11 |
| State | 0.94/0.49 | 0.55/0.08 | 0.85/0.58 | 0.66/0.21 |
| Symbol | 0.69/0.08 | 0.79/0.72 | 0.83/0.80 | 0.91/0.87 |

## K    DETAILS OF THE SECOND ORDER ANALYSIS

We now describe which automata properties we rely on in our second-order analysis. We refer to these as the **explanatory variables**. Depending on the intervention type, we use a variation of a set of explanatory variables for the WLS model. We use both local properties, those related specifically to the transitions or states we target, and global properties of the machine under scrutiny. The global properties are shared for all intervention categories, these are the expected length of strings generated by the machine we intervene on, and the machine's expected entropy. The local properties, on the other hand, differ between the intervention types. For the **transition interventions**, these are the entropy path-sum for the source state, the entropy path sum for the target state, the transition weight, the local source state entropy, and the target state entropy. In the case of **state interventions**, the local properties we consider are the entropy path sum for the state and the local entropy of the state. For **symbol interventions**, we only consider the machine's global properties.

The entropy path-sum of a state $q$ is the path-sum calculated over the machine we get from lifting the target machine such that the new weights are the entropy of the original machine, $-\log(w)$. The entropy of a given state is a measure of how even the weights of the outgoing transitions are. A higher entropy intuitively means it should be harder to model the state. The entropy path-sum then measures how distributed the probability mass over all substrings leading up to the state.

Specifics of the fitted WLS models are given in App. L.

## L    DETAILS OF FITTING WLS MODELS TO INTERVENTION TRENDS

We fit linear models to the trends over the sampled machines and then fit secondary weighted linear models to the coefficients of the first model. We provide some details of these models in the sections below, as well as an overview of the adjusted $R^2$ values in table Tab. 1.

### L.1    INTERCEPTS

Tables with information about the WLS fitted to the intercepts of the intervention trends are given in Tab. 2 (Transitions), Tab. 3 (States) and Tab. 4 (Symbols).

### L.2    SLOPES

Tables with information about the WLS fitted to the slopes of the intervention trends are given in Tab. 5 (transitions), Tab. 6 (states) and Tab. 7 (symbols).

## M    INTERVENTION TRENDS

We give some examples of the trends for the randomly sampled state interventions in Fig. 6b, and for the symbols in Fig. 7b. A corresponding figure for the transition interventions is given in Fig. 3b.

Table 2: Estimated coefficients ($\widehat{\beta}$), standard errors (SE), and $p$-values for a weighted linear model over the intercepts of the transition interventions.

| | Transformer | | | | | | RNN | | | | | |
| | KL | | | Decomp KL | | | KL | | | Decomp KL | | |
| Predictor | $\widehat{\beta}$ | SE | $p$-value | $\widehat{\beta}$ | SE | $p$ | $\widehat{\beta}$ | SE | $p$ | $\widehat{\beta}$ | SE | $p$-value |
|---|---|---|---|---|---|---|---|---|---|---|---|---|
| Intercept | 72.7 | 1.32 | 0.00 | 0.06 | 0.01 | 0.00 | 77.9 | 1.24 | 0.00 | 0.01 | 0.02 | 0.81 |
| Src. e.p.s. | -40.8 | 3.65 | 0.00 | 0.02 | 0.03 | 0.48 | -97.0 | 8.38 | 0.00 | -0.42 | 0.11 | 0.00 |
| Tgt. e.p.s. | -19.3 | 6.95 | 0.01 | -0.03 | 0.03 | 0.29 | -25.2 | 6.41 | 0.00 | 0.01 | 0.03 | 0.76 |
| Trans w. | 2.8 | 1.59 | 0.08 | 0.01 | 0.01 | 0.63 | -1.4 | 1.51 | 0.36 | 0.01 | 0.00 | 0.00 |
| Tgt. entr. | -1.6 | 1.22 | 0.18 | 0.00 | 0.00 | 0.25 | 1.4 | 0.71 | 0.05 | 0.00 | 0.00 | 0.39 |
| Src. entr. | 0.2 | 0.92 | 0.84 | 0.01 | 0.00 | 0.04 | 0.3 | 0.94 | 0.73 | -0.00 | 0.00 | 0.83 |
| Exp. len. | 83.4 | 28.15 | 0.00 | 0.12 | 0.08 | 0.15 | 99.6 | 39.62 | 0.01 | 0.08 | 0.06 | 0.19 |
| PFSA entr. | 1.0 | 29.86 | 0.97 | -0.13 | 0.09 | 0.19 | -4.3 | 37.74 | 0.91 | -0.05 | 0.06 | 0.40 |

Table 3: Estimated coefficients ($\widehat{\beta}$), standard errors (SE), and $p$-values for a weighted linear model over the intercepts of the state interventions.

| | Transformer | | | | | | RNN | | | | | |
| | KL | | | Decomp KL | | | KL | | | Decomp KL | | |
| Predictor | $\widehat{\beta}$ | SE | $p$-value | $\widehat{\beta}$ | SE | $p$ | $\widehat{\beta}$ | SE | $p$ | $\widehat{\beta}$ | SE | $p$-value |
|---|---|---|---|---|---|---|---|---|---|---|---|---|
| Intercept | 49.2 | 2.40 | 0.00 | 0.91 | 0.19 | 0.00 | 80.7 | 1.41 | 0.00 | 0.21 | 0.02 | 0.00 |
| FW entr. | -439.7 | 15.12 | 0.00 | 5.02 | 1.50 | 0.00 | -18.3 | 3.87 | 0.00 | 0.16 | 0.05 | 0.00 |
| Local entr. | -1.5 | 1.52 | 0.33 | 0.01 | 0.01 | 0.16 | -1.1 | 1.04 | 0.28 | -0.02 | 0.01 | 0.00 |
| Exp. len. | 258.2 | 229.16 | 0.26 | -2.65 | 1.38 | 0.06 | 17.0 | 29.73 | 0.57 | -0.05 | 0.07 | 0.45 |
| PFSA entr. | -39.4 | 231.27 | 0.86 | 2.84 | 1.35 | 0.04 | 34.9 | 29.34 | 0.24 | 0.05 | 0.08 | 0.52 |

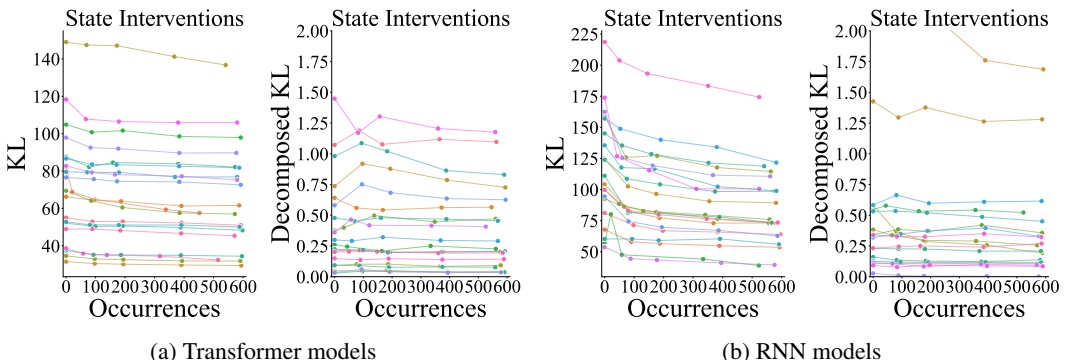

(a) Transformer models
(b) RNN models

Figure 6: A subset of state intervention trends.

Table 4: Estimated coefficients ($\widehat{\beta}$), standard errors (SE), and $p$-values for a weighted linear model over the intercepts of the symbol interventions.

| | Transformer | | | | | | RNN | | | | | |
| | KL | | | Decomp KL | | | KL | | | Decomp KL | | |
| Predictor | $\widehat{\beta}$ | SE | $p$-value | $\widehat{\beta}$ | SE | $p$ | $\widehat{\beta}$ | SE | $p$ | $\widehat{\beta}$ | SE | $p$-value |
|---|---|---|---|---|---|---|---|---|---|---|---|---|
| Intercept | 66.1 | 1.18 | 0.00 | 1.60 | 0.03 | 0.00 | 89.3 | 1.28 | 0.00 | 2.36 | 0.04 | 0.00 |
| Exp. sym. freq. | 2.9 | 0.87 | 0.00 | 0.28 | 0.03 | 0.00 | 1.2 | 0.80 | 0.13 | 0.18 | 0.02 | 0.00 |
| Exp. len. | 81.6 | 16.49 | 0.00 | 1.25 | 0.34 | 0.00 | 8.1 | 6.30 | 0.20 | 3.63 | 0.58 | 0.00 |
| PFSA entr. | -55.2 | 16.52 | 0.00 | -0.39 | 0.34 | 0.26 | 62.4 | 4.58 | 0.00 | -0.04 | 0.36 | 0.91 |

Table 5: Estimated coefficients ($\widehat{\beta}$), standard errors (SE), and $p$-values for a weighted linear model over the slopes of the transition interventions.

| | Transformer | | | | | | RNN | | | | | |
| | KL | | | Decomp KL | | | KL | | | Decomp KL | | |
| Predictor | $\widehat{\beta}$ | SE | $p$-value | $\widehat{\beta}$ | SE | $p$ | $\widehat{\beta}$ | SE | $p$ | $\widehat{\beta}$ | SE | $p$-value |
|---|---|---|---|---|---|---|---|---|---|---|---|---|
| Intercept | -0.007 | 0.000 | 0.000 | -0.000 | 0.000 | 0.000 | -0.013 | 0.000 | 0.000 | -0.000 | 0.000 | 0.030 |
| Src. e.p.s. | -0.004 | 0.001 | 0.000 | 0.000 | 0.000 | 0.073 | 0.005 | 0.003 | 0.090 | -0.000 | 0.000 | 0.724 |
| Tgt. e.p.s. | -0.004 | 0.002 | 0.066 | -0.000 | 0.000 | 0.001 | 0.002 | 0.004 | 0.007 | -0.000 | 0.000 | 0.780 |
| Trans w. | 0.001 | 0.000 | 0.011 | -0.000 | 0.000 | 0.386 | 0.001 | 0.001 | 0.395 | -0.000 | 0.000 | 0.487 |
| Tgt. entr. | -0.000 | 0.000 | 0.245 | -0.000 | 0.000 | 0.001 | -0.001 | 0.000 | 0.014 | 0.000 | 0.000 | 0.690 |
| Src. entr. | 0.001 | 0.000 | 0.091 | -0.000 | 0.000 | 0.000 | -0.001 | 0.000 | 0.047 | -0.000 | 0.000 | 0.427 |
| Exp. len. | 0.006 | 0.008 | 0.432 | 0.000 | 0.000 | 0.267 | 0.003 | 0.015 | 0.828 | 0.000 | 0.000 | 0.604 |
| PFSA entr. | -0.006 | 0.009 | 0.475 | -0.000 | 0.000 | 0.459 | -0.018 | 0.014 | 0.210 | -0.000 | 0.000 | 0.626 |

Table 6: Estimated coefficients ($\widehat{\beta}$), standard errors (SE), and $p$-values for a weighted linear model over the slopes of the state interventions.

| | Transformer | | | | | | RNN | | | | | |
| | KL | | | Decomp KL | | | KL | | | Decomp KL | | |
| Predictor | $\widehat{\beta}$ | SE | $p$-value | $\widehat{\beta}$ | SE | $p$ | $\widehat{\beta}$ | SE | $p$ | $\widehat{\beta}$ | SE | $p$-value |
|---|---|---|---|---|---|---|---|---|---|---|---|---|
| Intercept | -0.006 | 0.001 | 0.000 | -0.000 | 0.000 | 0.095 | -0.013 | 0.000 | 0.000 | -0.000 | 0.000 | 0.000 |
| FW entr. | 0.004 | 0.004 | 0.240 | -0.001 | 0.000 | 0.280 | 0.003 | 0.001 | 0.007 | -0.000 | 0.000 | 0.086 |
| Local entr. | 0.001 | 0.000 | 0.002 | -0.000 | 0.000 | 0.592 | -0.000 | 0.000 | 0.559 | -0.000 | 0.000 | 0.311 |
| Exp. len. | 0.015 | 0.057 | 0.791 | -0.000 | 0.000 | 0.618 | -0.001 | 0.009 | 0.922 | -0.000 | 0.000 | 0.761 |
| PFSA entr. | -0.023 | 0.058 | 0.692 | 0.000 | 0.000 | 0.651 | -0.007 | 0.009 | 0.411 | 0.000 | 0.000 | 0.638 |

Table 7: Estimated coefficients ($\widehat{\beta}$), standard errors (SE), and $p$-values for a weighted linear model over the slopes of the symbol interventions.

| | Transformer | | | | | | RNN | | | | | |
| | KL | | | Decomp KL | | | KL | | | Decomp KL | | |
| Predictor | $\widehat{\beta}$ | SE | $p$-value | $\widehat{\beta}$ | SE | $p$ | $\widehat{\beta}$ | SE | $p$ | $\widehat{\beta}$ | SE | $p$-value |
|---|---|---|---|---|---|---|---|---|---|---|---|---|
| Intercept | -0.004 | 0.000 | 0.000 | -0.001 | 0.000 | 0.000 | -0.014 | 0.000 | 0.000 | -0.001 | 0.000 | 0.000 |
| Exp. sym. freq. | 0.000 | 0.000 | 0.002 | -0.000 | 0.000 | 0.000 | -0.001 | 0.000 | 0.000 | -0.000 | 0.000 | 0.722 |
| Exp. len. | -0.000 | 0.002 | 0.874 | -0.000 | 0.000 | 0.110 | -0.001 | 0.001 | 0.545 | -0.001 | 0.000 | 0.000 |
| PFSA entr. | -0.000 | 0.002 | 0.959 | -0.000 | 0.000 | 0.667 | -0.009 | 0.001 | 0.000 | -0.000 | 0.000 | 0.034 |

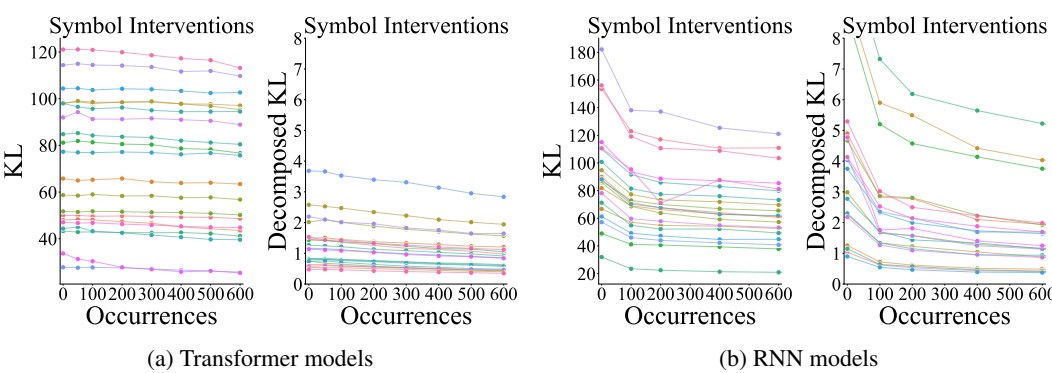

(a) Transformer models        (b) RNN models

Figure 7: A subset of symbol intervention trends.

