# OpenReview forum: "A Causal Study on The Learnability of Formal Languages"
_ICLR.cc/2025/Conference — Submitted to ICLR 2025_

### Official Review · Reviewer_FZWr · 2024-10-23

**Soundness:** 3
**Presentation:** 1
**Contribution:** 3
**Rating:** 6
**Confidence:** 3

**Summary:**

Previous works have studied language models by characterizing their performance on synthetic datasets samples from families of formal languages. This paper tries to get a more fine-grained understanding of learning language models on regular languages, by controlling for certain statistical features such as the number of times a state occurs. To achieve this, the authors introduce the counting semiring, which allows sampling from the finite state machine conditioned on the count of a feature of interest (e.g. the number of state transitions). Finally, this approach is applied to transformers and LSTMs to study what factors (causally) impact the learned models.

**Strengths:**

Overall, I like the direction this work takes; providing more fine-grained control in studying language models. As language models are often blamed for being overly reliant on the presence of statistical features, it is especially useful that the authors can correct for these and measure their impact. The proposed sampling approach seems well thought out and novel to me, although I must say I’m not that familiar with this area and so could very much be unaware of related work. Finally, the proposed approach is quite general, and hence might be of broader use outside of the specific demonstration given in the experimental section.

**Weaknesses:**

My main gripe with the paper lies in the presentation. Firstly, the paper does not include examples of the central contribution: the counting semiring. Secondly, the notation at many points feels cumbersome and was sometimes confusing. For example, $p(\pi)$ could both mean a probability of a path (e.g. proof of Thm3.2.) and the origin of a path (Eq. 28). For more examples, see the questions. I also felt that this was partly caused by trying to be very general. Indeed, the counting semiring is defined over any semiring, while it seems that in practice only the probabilistic semiring is ever used/needed.
Thirdly, the paper pushes some definitions to the Appendix (e.g. for forward and backward probability), making the paper less self-contained. On the other hand, rather straightforward proofs (e.g. for Thm 3.1) are kept in the paper.

Finally, the new insights provided by using the proposed counting semiring in the experimental section are quite limited. Perhaps this is subjective, but e.g. that transformers perform better than LSTMs is well known. The findings also do not appear to be very actionable.

**Questions:**

- I initially found the naming of "counting semiring" confusing, as it has been used before to describe the semiring over the natural numbers (which can be used to count the number of parses a word has). See e.g. the seminal semiring parsing work by Goodman et al. (1999).

- On a related note, the paper claims its counting semiring is novel, but it is essentially the same as a polynomial semiring with 1 variable (except that the convolution gets truncated), which has been well studied. I suppose the two pages of proofs in Appendix C wouldn’t be necessary if you observe that there is a homomorphism from the polynomial into the counting semiring.

- Def. 3.3. is unclear. Using both a feature function $\phi$, a set of features $\Phi$, and a lifting function $\mathcal{L}$ seems to unnecessarily complicate things. Furthermore, $\phi$ is a function in $\delta \to \{ 0, 1,\dots N \}$, but then the lifting function uses $\phi(\alpha)$ where $\alpha \in \mathbb{K}$. More importantly, Def 3.3 also doesn’t explicitly define the counting automaton! I assume the transition weights in the counting automaton should contain the one-hot vectors of the lifting function times the original weights, but this is not mentioned.

- The proof of Thm 3.2. seems flawed. I would have expected something like $\sum_{\lvert \pi \rvert_\Phi = n} p(\pi) = \sum w_I (\pi)_n$. Could you elaborate on how the step in Eq. uses Thm 3.1? Why don’t you need a subscript for $w_i(\pi)$ (i.e. you suddenly go from a probability to a vector)? Also: why is the normalization with $\sum p(\pi)$ necessary, isn’t this equal to 1?

- In Corollary 4.1, there’s no mention of what v stands for.

_Minor comments_

- On line 129, What does the \cdot stand for?

- In Figure 3, should the transitions be a_1 and a_2 instead of a and b? (As line 707 defines $\Sigma = \{ a_1, a_2 \}$). I suppose that w_1 and w_2 in this example also should both be 1, instead of arbitrary numbers, as the WFSA is mentioned to be probabilistic)

- On line 806, there is a broken reference.

**References**

Goodman, J. (1999). Semiring parsing. *Computational Linguistics*, *25*(4), 573-606.

---

> ### Author Response · Authors · 2024-11-13
> **Quick question**
>
> Thanks so much for the detailed review! We will digest all the comments over the week and make substantial changes to the presentation in the manuscript. Many of the comments will certainly improve our work.
>
> One initial thought is about your remark “On a related note, the paper claims its counting semiring is novel, but it is essentially the same as a polynomial semiring with 1 variable (except that the convolution gets truncated), which has been well studied. I suppose the two pages of proofs in Appendix C wouldn’t be necessary if you observe that there is a homomorphism from the polynomial into the counting semiring.”
>
> It’s true that we can construct an isomorphism to the polynomial semiring over 1 variable with truncated convolution as the times. And, it’s true the two pages of proof are routine with one exception. That exception is the construction of the Kleene star and an algorithm for it. Goodman (1999) does not deal with the Kleene star. Indeed, deriving the Kleene star was the only non-trivial part of the proof in our eyes. Do you know a reference for the claim that this exists in the literature? We would happily replace the tedious appendix with a citation, but we had to work out the details ourselves so we are working towards a publication that includes it for completeness of the literature.

---

> > ### Comment · Reviewer_FZWr · 2024-11-15
> >
> > My comment was about the proof that a polynomial semiring is a semiring. Not the other stuff like proving that the polynomial of a closed semiring is again closed. In general, my goal here is definitely not to argue against having explicit proofs, just that the counting semiring could have been connected more with existing constructions. Most existing literature on Kleene stars on semirings assumes idempotent semirings, so it indeed might be difficult to find clean-cut references for e.g. Prop D.1.

---

> > > ### Author Response · Authors · 2024-11-17
> > > **thanks**
> > >
> > > Thanks! Do you have a reference for a truncated polynomial semiring? We are trying to figure out if we can indeed shorten the proof for the properties other than the Kleene star.

---

> > > > ### Comment · Reviewer_FZWr · 2024-11-18
> > > >
> > > > I'm not a mathematician, so I'm perhaps not the best person to ask, but I've looked up some references to hopefully point you in the right direction.
> > > >
> > > > You can see truncation as defining a congruence class on the polynomial semiring. Similar to quotient rings, these give rise to a new semiring (see e.g. chapter 8 on factor semirings in the seminal semiring book by Golan: "Semirings and their applications"). In other words, your counting semiring on a base semiring $S$ equals the semiring $S[x]/x^{n+1}$.
> > > >
> > > > For references on Kleene stars of semirings, a good place to start looking might be chapter 2 in the PhD thesis "The Semiring-Based Provenance Framework for Graph Databases" by Ramusat.

---

> ### Author Response · Authors · 2024-11-18
> **thanks**
>
> Thanks! We did some digging. I think the following construction would work. I think the semiring of truncated polynomials is in fact a ring. Here's a sketch of the derivation. Let R[x] be the ring of polynomials. Suppose we want truncated at $N$. For the ring of polynomials, all polynomials with degree > $N$ form an ideal. Call this $I$. Then, we can compute the quotient ring $R[x] / I$. I think this quotient ring is isomorphic to the truncated semiring. This, of course, implies we will get an inverse. We have to push this through.
>
> With respect to your suggestion, it's true that the monic polynomials $a x^{i+1}$ form an ideal and, thus, we could construct $R[x] /
> x^{i+1}$, but this would not be isomorphic to our truncated ring. I think the above generalization can be made to be pushed through, though.
>
> Do you think this derivation would help the paper? It may be shorter, as you suggest, but it requires more knowledge of algebra. Maybe we could provide both?
>
> Re the Kleene star, I think the provenance semiring is not close enough to what we want. As I understand it, it's very similar to Goodman's derivation semiring. We will keep searching.
>
> EDIT: We actually now think the provenance semiring framework might be general enough to handle our use case. We will keep you posted.
>
> One final question: What would you think an appropriate claim with respect to novelty is?

---

> > ### Author Response · Authors · 2024-11-19
> > **Other comments**
> >
> > Thanks again for the detailed and helpful feedback! We thought we’d answer the other questions and concerns while working on the updated manuscript:
> >
> > **1. Examples of the counting semiring are missing**
> >
> > This was a blunder, we will include some more detailed examples.
> >
> > **2. The notation at many points feels cumbersome and was sometimes confusing.**
> >
> > We agree and will simplify to fix the issues you point out.
> >
> > **3. The counting semiring is defined generally but in practice only needed over the probabilistic semiring.**
> >
> > We do see some merit in defining it generally. In practice, we use the counting semiring over the log-semiring to improve numerical stability. We will comment on this in the updated version.
> >
> > **4. Definitions in the appendix should be kept in the main text.**
> >
> > Good point, we will modify the structure to make the work more self-contained.
> >
> > **5. Finally, the new insights provided by using the proposed counting semiring in the experimental section are limited. Perhaps this is subjective, but e.g. that transformers perform better than LSTMs is well known. The findings also do not appear to be very actionable.**
> >
> > We agree that it is known that Transformers perform better than LSTMs in general and that the findings are not very actionable as such. The experiments are conducted as a demonstration of how sampling with the semiring can be used. By sampling under the interventions, we can track how local structural properties impact the learnability of transitions, states, and symbols.
> >
> > **6. The name “counting semiring” has been used before**
> >
> > We were not aware of this and think the naming conflict is unfortunate, we have some new candidates in mind (marginal/binning/bucket/occurrence).
> >
> > **7. Def. 3.3. is unclear. An explicit definition of the counting automaton should contain one-hot vectors.**
> >
> > Thanks, we agree it can be simplified. And the explicit definition certainly makes it better, there is an attempt at this in line 180 but it is missing the multiplication with the transition weight.
> >
> > **8. Thm 3.2:  Why don’t you need a subscript for the inner path weight (i.e. you suddenly go from a probability to a vector)? Could you elaborate on how it uses Thm 3.1? Why is the normalization necessary?**
> >
> > We see the issues, the way it is currently written we thought of $\pi$ as a path in the original automaton.  So the probability of the path is the same as the inner path weight and no subscript is needed. Normalization is not needed since we have probabilities already (the denominator is always 1).  Instead of $w_{I}(\pi)$ we can write $w_{I}(\pi)_n$ for the vector in the lifted machine. The point with referencing Thm 3.1 is that it tells us the probability is the n-th value of the inner weight, and then we no longer need the specific subscript on the sum as you point out, we’ll make this clearer in Thm 3.1. This was very helpful.
> >
> > **9. In Corollary 4.1, there’s no mention of what v stands for.**
> >
> > Thanks, we will fix this, it’s the transition weight.
> >
> >
> > **10. On line 129, what does the \cdot stand for?**
> >
> > Thanks, this should have been $\otimes$, we will fix this.
> >
> > **11. In Figure 3, should the transitions be a_1 and a_2 instead of a and b? (As line 707 defines ). I suppose that w_1 and w_2 in this example also should both be 1, instead of arbitrary numbers, as the WFSA is mentioned to be probabilistic**
> >
> > Yes on both comments, we’ll add the sum-to-1 constraint and fix the figure.

---

> > > ### Author Response · Authors · 2024-11-28
> > > **update on isomorpishm**
> > >
> > > We have updated the pdf and have included a proof of the isomorphism of the $N$-th order semiring (which we now refer to as the marginal semiring) to the truncated polynomial semiring over $x^{N+1}$ (Theorem G.1).
> > >
> > > The equivalence could save some ink in the appendix, but we leave the derivations in with a remark for completeness. While it is now clear that the marginal semiring has this correspondence, it still remains a key actor in our facilitation of the causal study of the learnability of formal languages---something that to the best of our knowledge has not been done before. This includes the sampling procedures, and how they facilitate causal analysis of neural networks learnability of targeted properties using the decomposed KL divergence.
> > >
> > > Thanks for pointing us in this exciting direction, we are still thinking about what other implications this has.

---

> > > > ### Comment · Reviewer_FZWr · 2024-11-28
> > > >
> > > > I'm happy to see the authors have been further exploring their work.
> > > >
> > > > > We have updated the pdf and have included a proof of the isomorphism of the N-th order semiring
> > > >
> > > > I don't think that the new Theorem G.1. is correct. A convolution of $\mathbb{R}[x] / x^N$ doesn't truncate but wraps around. E.g. for $N=3$, we get $x^2 \otimes x^2 = x$ and not $x^2 \otimes x^2 =0$. This is why I was talking about congruence classes before.
> > > >
> > > > > While it is now clear that the marginal semiring has this correspondence, it still remains a key actor in our facilitation of the causal study of the learnability of formal languages---something that to the best of our knowledge has not been done before.
> > > >
> > > > I agree that whether your semiring construction is novel or not isn't crucial here.

---

> > > > > ### Author Response · Authors · 2024-11-28
> > > > >
> > > > > We think there is a small misunderstanding in your counterexample.
> > > > >
> > > > > The quotient ring is defined such that all elements in the ideal null out, this means that any multiple of $x^{n+1}$ is zeroed, including all terms with $x^m$ such that $m\geq n+1$. The ideal $I$ of a ring $R$ is a subset that (to quote Wikipedia) "absorbs multiplication from the elements of R", defined such that $(I, +)$ is a subgroup of $(R, +)$ and for every $r\in R$ and every $x\in I$ then $rx\in I$. This means that $x^4\in I$ if $I$ is the ideal spanned by all polynomials of degree 3. So there is no wrapping around, it's all absorbed.

---

> > > > > > ### Comment · Reviewer_FZWr · 2024-11-28
> > > > > >
> > > > > > I verified it and you are right. My apologies, I was going too fast and am very rusty on this material.
> > > > > >
> > > > > > I changed my score as I believe the authors have improved the presentation of the paper and provided a better analysis of the results.

---

> > > > > > > ### Author Response · Authors · 2024-11-28
> > > > > > >
> > > > > > > Thanks a lot!

---

### Official Review · Reviewer_13Db · 2024-11-02

**Soundness:** 4
**Presentation:** 2
**Contribution:** 2
**Rating:** 5
**Confidence:** 2

**Summary:**

This paper proposes a efficient method for sampling strings from a probabilistic finite state automata (PFSA) subject to certain constraints. The main application of the technique is in computing the causal effect of certain dataset properties (such as symbol frequencies) on different probabilistic language models (specifically, Transformers and LSTMs).

Preliminary experiments are conducted using PFSA with 100 states and 10 symbols. Experiments explore 3 properties: increasing the number of times a given transition (74 machines), state (149 machines), or symbol (73 machines) occurs when sampling a dataset from PFSA. Two types of analysis are performed. The first is a linear regression of how frequently a given property occurs against the KL divergence between the PFSA and LM output distribution. The second is an analysis of how various properties of the underlying PFSA (e.g., expected string length) relates to the coefficients of the fitted regression.

===

I have read the author's responses. While they have addressed some of my concerns (e.g., length generalization being another benefit of their sampling algorithm), my main concern is still that the analysis of results is rather limited in scope to Transformers / RNNs trained on strings sampled from PFSA, which limits the contributions. I also note that the authors have not answered my question about a full derivation of the runtime complexities, which I think is key given that the improved runtime is a major claimed benefit of the approach.

I thus maintain my score

===

After additional discussion I have raised my score

**Strengths:**

The idea to explore how different properties of a data generation process are causally related to the trained performance of different LM architectures (transformer, LSTM) is an important topic toward understanding the behavior of LLMs.

The sampling algorithm is, to the best of my knowledge, original, and asymptotically improves upon the naive approach of intersecting FSAs.

The results are also interesting, although I found it difficult to grasp the significance of the results given the specificity of the domain (language modeling applied to strings sampled from a PFSA).

**Weaknesses:**

The majority of the paper is dedicated to developing an extremely general and abstract framework for efficient constrained generation from PFSA. However, I view the significance of the generality to be somewhat low, given that the experiments are conducted in a single, much more specific setting. For instance, I don't think much is gained by defining the algorithm in terms of a counting semi-ring instead of directly describing the data structure used to track feature counts. I think the presentation would be clearer by directly presenting the specific version of the algorithm actually used in the experiments in the main text, and developing the more general theory in the appendix. This would also free up space for a deeper analysis of the experimental results.

The technique is also limited to finite state automata. Modeling complex joint distributions with FSA incurs a large computational overhead, so it's unclear how to apply a technique based on FSA (even with the asymptotic improvements of the new sampling algorithm) to study more complex types of dataset properties or even natural language.

Finally, I found the results and discussion to be somewhat limited. One suggestion would be to contextualize the observed results to explain differences observed between Transformers and LSTMs in other domains (such as natural language).

More minor concerns:

- The notation gets unclear at points. For instance, $n_i$ is not defined on line 93.
- There are missing references [1] [2]

[1] Allen-Zhu, Zeyuan, and Yuanzhi Li. "Physics of language models: Part 1, context-free grammar." arXiv preprint arXiv:2305.13673 (2023).

[2] Jin, Charles. "Latent Causal Probing: A Formal Perspective on Probing with Causal Models of Data." arXiv preprint arXiv:2407.13765 (2024).

**Questions:**

Can you provide derivations for the runtime complexities in lines 125-128?

Is there another instance of a counting semi-ring that would benefit from the sampling algorithm (beyond string feature counts?)

---

> ### Author Response · Authors · 2024-11-19
>
> We are glad you found the work interesting and will address the unclear notation.
>
> **1. The majority of the paper is dedicated to developing an extremely general and abstract framework for efficient constrained generation from PFSA. I don't think much is gained by defining the algorithm in terms of a counting semi-ring instead of directly describing the data structure used to track feature counts. More general theory should be put in the appendix. This would also free up space for a deeper analysis of the experimental results.**
>
> We agree that the paper is a bit heavy on the abstract and general, and will make some changes to make the experimental section more prominent. We see the general nature of the proposed theory as a strength but we agree that we should make the practical benefits clearer, see for instance our first response to reviewer FMxv. If we had described data structures directly, i.e. only in the context of the experiments it would have been hard to derive the same general results that can be applied to other situations or as easily extended to more complex pattern counting.
>
> **2. Modeling complex joint distributions with FSA incurs a large computational overhead, so it's unclear how to apply a technique based on FSA (even with the asymptotic improvements of the new sampling algorithm) to study more complex types of dataset properties or even natural language.**
>
> We don’t see the computational overhead of using synthetic data as a tradeoff, the synthetic languages enable us to study properties that we could not study using natural language—since we can’t control the generative process in the same way. Synthetic methods also enable sampling of an arbitrary amount of languages, facilitating more robust analysis, rather than focusing on specific languages. We cite a fair amount of related literature on this topic and will clarify this point in the writing. We do however acknowledge the importance of studying more complex datasets and those inspired by natural language. We see the suggested methodology as a key part of doing this, e.g. by (1) scaling up language complexity using strongly connected components with Tarjans’s (or Kosaraju’s) algorithm, or (2) analyzing properties of natural languages through formal grammars that are defined over parts of them, by converting these to simpler machines the semiring can be used for controlled sampling and analysis of structural properties that relate to natural language.
>
> **3. One suggestion would be to contextualize the observed results to explain differences observed between Transformers and LSTMs in other domains (such as natural language).**
>
> This is a good suggestion, we will expand on this in the updated version over the coming days.

---

> ### Author Response · Authors · 2024-11-27
> **Complexity analysis and further experiment**
>
> Thanks for the update in the summary, let us answer your concerns more directly.
>
> **Runtime complexities**
>
> We describe the complexity of $v^*$, and the use of the fast-fourier theorem, in the appendix in lines 1022-1043. But this should all have been made clearer and fully contextualized as you point out. The cost of the convolution would have been $O(n^2)$ since we do the scalar multiplication $1+2+3 + … + (n+1) = (n+1)(n+2)/2=O(n^2)$ times, but the FFT trick brings this down to $O(n\cdot \text{log})(n)$. Now for the efficiency of the sampling:
>
> (1) Occurrence sampling: We first sample the number of occurrences of the property we target, i.e. how often the property should occur in each of the $K$ strings. We need to do $K$ convolutions each with the cost of a convolution, $n\text{log}(n)$, giving us $K\cdot n\text{log}(n)$. Even if we store the prior result for practical gains this does not improve the big-O.
>
> (2) Property occurrence sampling: We first need to calculate the pathsum (allsum) of all states. The pathsum computation using Lehmann’s algorithm requires $O(|Q|^3)$ operations (since we need to do $|Q|$ iterations in the calculations for $|Q|$ states and max $|Q|$ transitions in a fully connected graph). Then for each of these operations, we need to do the multiplication over a vector of size $n+1$, which we can do in $n \text{log}(n)$ using the FFT approach. So we get $O(|Q|^3n \text{log}(n))$. Then we sample a symbol for each step in the max length, let’s call this $L$, and we have $K$ strings for which we need to do a convolution each so we get $O(KLn\text{log}(n))$. This means the pathsum calculations dominate unless $KL>|Q|^3$.
>
> Also, refer to our response to FMxv about the complexity of the prior approach.
>
> **Additional focus on experiments**
>
> While the experimental discussion was a bit sparse, our focus was on demonstrating what can be done with controlled targeted sampling using the semiring. We are, however, also adding new experiments, where we demonstrate directly the causal impact of frequencies of symbols on their learnability. We do this by causally sampling under the counting intervention, we then compare the results to an ancestral sampling baseline. We plot the Monte Carlo estimate of the expected decomposed KL for the symbol transitions against the number of occurrences in the training corpus. See https://i.imgur.com/1jfDXPK.png for a plot comparing the two (updated causal graph at https://i.imgur.com/MBpjt0h.png). It shows that the causal interventions indicate the impact of the number of occurrences is overestimated for low occurrences. At the same time, it is underestimated for higher frequencies, compared to relying on standard sampling. This demonstrates how our method can be used to analyze direct causal effects given an intervention on the property count in a PFSA—something that has not been done before to the best of our knowledge. We maintain that the key contribution is the mechanism for controlled sampling, and how it enables causal analysis like the one we just described, not the comparison between RNNs and Transformers as such--we will do our best to clarify this in the manuscript.
>
> **Aditional use-cases beyond string feature counts**
>
> The semiring and sampling algorithm could also be applied to different settings than analyzing counts of symbols in strings. Is this what you are asking about? That is, as long as the process can be modeled by a probabilistic automaton. Consider for instance, as an example, if one wants to construct a playlist and there are probabilities of what song a person likes given another song, then we might want to constrain the number of songs included by a specific artist, the counting intervention would allow this.
>
> ---
>
> We have also checked the citations you recommend—these are relevant and we’ll incorporate them!
>
> Are you OK with the complexity analysis? We are happy to provide more details and are grateful for any comments or suggestions you think can improve the clarity.

---

> ### Author Response · Authors · 2024-12-01
>
> As the discussion period is ending soon, we would like to kindly check if our runtime complexity analysis addresses your concerns. We understand that reviewing is a time-consuming process, and we greatly appreciate the effort you’ve already put into evaluating our work. If there are any further questions or clarifications needed, we’d be happy to address them.

---

> > ### Comment · Reviewer_13Db · 2024-12-01
> > **Response**
> >
> > Thanks, I appreciate the explicit complexity analysis.
> >
> > One additional suggestion as I read the updated manuscript - it would be good to have wall-clock times for how long sampling with the marginal semi-ring takes vs. the naive approaches of rejection sampling or intersection the FSA.

---

> > > ### Comment · Reviewer_13Db · 2024-12-01
> > > **Clarification on new results (Section 7)**
> > >
> > > Can you clarify what the exact experimental set up is in the (new) Section 7? Am I correct in understanding
> > >
> > > - Pick a single symbol
> > > - Sample 400 machines
> > > - Use ancestral sampling to sample 500 strings from each of the 400 machines, producing 400 datasets of 500 strings each
> > > - Use "causal" sampling to sample 500 strings from each of the 400 machines, producing 400 datasets of 500 strings each, where the intervention is $count(symbol) \in \\{100, 200, \dots, 900 \\}$.
> > > - Train an LM (transformer?) on each of the datasets, producing 400 LMs trained on the ancestral datasets, and 400 trained on the causal datasets.
> > > - Estimate the decomposed KL divergence of each LM against the machine used to produce its training data
> > > - Plot each KL divergence against the empirical frequency of the symbol in the training data
> > >
> > > Questions
> > > - Does ancestral sampling produce the distribution as "causal" sampling except without any interventions?
> > > - I would have expected there to be far more samples falling in the middle (around 500) rather than at the ends (100 and 900) for ancestral sampling, so why is error roughly the same for all counts? Also, why would the error bars be tighter for ancestral sampling instead of causal sampling?
> > > - Do you have any intuitions for why the observed results differ?

---

> > > > ### Author Response · Authors · 2024-12-01
> > > >
> > > > Thank you for your thoughtful comments. Your understanding is mostly correct, and we recognize the need to clarify certain aspects of the setup and results. In the intervention sampling, we targeted a single symbol. For non-intervention sampling, we used all symbols to compute decomposed KL divergences for the entire vocabulary. This increased the data points available for the ancestral setup, which impacts the difference in error bars. For section 7 we only trained LSTM RNNs. We will clarify this and make sure to compare both LSTMs and Transformers for the final version.
> > > >
> > > > Here are responses to your specific questions:
> > > >
> > > > **Does ancestral sampling produce the distribution as "causal" sampling except without any interventions?**
> > > >
> > > > Yes, this is a key feature of the presented method.
> > > >
> > > > **I would have expected there to be far more samples falling in the middle (around 500) rather than at the ends (100 and 900) for ancestral sampling, so why is the error roughly the same for all counts?**
> > > >
> > > > For the intervened samples, the distribution is controlled, leading to similar error behavior across counts. For the ancestral samples, while there are indeed more data points in the middle range (as shown in the plot https://i.imgur.com/qbcWZZv.png), the large number of overall points reduces variability, and this uniformity in sampling might result in comparable error across counts.
> > > >
> > > > **Also, why would the error bars be tighter for ancestral sampling instead of causal sampling?**
> > > >
> > > > The tighter error bars in ancestral sampling are due to the larger number of data points available, as we use the full vocabulary rather than targeting specific symbols.
> > > >
> > > > **Do you have any intuitions for why the observed results differ?**
> > > >
> > > > The observed differences stem from the interplay between symbol frequency and the joint distribution of symbols in the language. For ancestral sampling, relationships between symbols naturally reflect their co-occurrence patterns in the joint distribution. In contrast, intervention-based sampling alters a symbol's frequency while leaving its co-occurrences unchanged, potentially distorting learned dependencies. This hypothesis aligns with our results but warrants further investigation to understand the full impact on learning dynamics.
> > > >
> > > >
> > > > ----
> > > > We are working on obtaining wallclock comparisons between rejection sampling and our method.

---

> ### Comment · Reviewer_13Db · 2024-12-01
>
> Thanks for the clarifications. Echoing Reviewer H83n's concerns I do think that the writing could be a bit more clear on how exactly the paper's contributions relate to "causality". Based on Section 7 and the rewrite, my understanding is that this paper is arguing for more studies that train LMs on different datasets to study the (causal) effect of the dataset on the trained LM, rather than studying the behavior of a single (pretrained) LM. To that end, the paper's main contribution is a more efficient way to sample datasets with certain constraints from FSAs.
>
> Similar to Reviewer FMxv I think the paper could use additional motivation for why the algorithm represents a major advancement in this line of research. For instance, I think the motivation would be more convincing if the authors could point a number of pressing experiments that are just waiting for an efficiently sampling method. However, in my opinion, the reality is that designing interesting, well-motivated interventions is itself the more salient question (rather than being able to sample from the posterior efficiently).
>
> I have therefore revised my score up to a 5, but will not object if the paper is ultimately accepted.
>
> Given that we are reaching the end of the discussion period, I will leave the following suggestions for the authors for future revisions.
> - Lines 37-39 are confusing: "Sampling from formal languages allows us to correlate properties of language with the performance of different architectures", but then none of the citations involve sampling from formal languages, but are rather studies of pretrained LLMs. I think this line is trying to draw the distinction described in my first paragraph (causal studies that train LMs on different datasets vs. observational studies that study a pretrained LM). I might suggest including an explicit example of an observational study to help drive the distinction home.
> - I would suggest including wall-clock times for FSA intersection as an additional baseline (to rejection sampling). My main question is whether the included experiments are feasible without the techniques developed in the paper.
> - The citation [1] (and possibly its predecessor [2]) should be included in the related works for causal analysis (e.g., line 40).
>
> [1] https://openreview.net/forum?id=98ekcwQqb7
>
> [2] https://proceedings.mlr.press/v235/jin24e.html

---

> ### Author Response · Authors · 2024-12-02
>
> Thanks for increasing the score!
>
> Regarding H83n’s concern about causality, we appreciate their acknowledgment that the revision improved clarity. To summarize, our work introduces a method for efficient controlled sampling from PFSAs, enabling systematic interventions to study the causal factors affecting language models' learnability. The derivation of the decomposed KL-divergence provides a direct measure of these effects, marking a key step forward in causal analysis.
>
> Our primary focus is on introducing a methodology for training multiple LMs on diverse datasets to causally analyze the learnability of different architectures. This approach sheds light on what properties specific architectures can learn and generalize. While this work does not extend to causal studies of pre-trained general-purpose models, it establishes a foundation for such investigations in the future, particularly for analyzing in-context learning.
>
> On the significance of our algorithm: we agree that designing well-motivated experiments is important. At the same time, we stress the value of providing the methodological tools to make such experiments feasible. For instance, our method supports Monte Carlo sampling over a family of automata, enabling causal exploration of generalization at scale. Examples include: How does training on strings up to a certain length affect the learnability of longer strings? Or how does training on data from one generative structure transfer to another overlapping structure (e.g., generalizing from 'run' to 'jump')?
>
> We appreciate the feedback on lines 37-39 and the suggestion of additional references; these will be addressed in revision.
> Lastly, while the complexity of our approach is established, we are actively working to include wall-clock times for FSA intersection as a baseline to further demonstrate feasibility.

---

> > ### Author Response · Authors · 2024-12-02
> >
> > Apologies for the confusion. Upon further consideration, we have not found any implementation or writeup that uses intersection in the same way as in our approach. While it is straightforward (at least for symbols) to intersect a machine that tracks a certain number of occurrences with a randomly sampled automaton, we are not aware of any prior method for sampling many strings such that the overall property being targeted achieves a specific cumulative number of occurrences (as formalized in Theorem 4.2 of our work).
> >
> > As such, we focus our comparison on the efficiency of rejection sampling versus our method. We share some initial wall-clock time results below and will elaborate further in the final version.
> >
> > We sampled machines with 10 symbols and 100 states. We then generated 1000 strings with 100 occurrences of a randomly chosen symbol. This experiment was repeated for 10 different machines. On our hardware, rejection sampling took an average of 10 minutes to sample 1000 strings, with a worst-case runtime of 22 minutes. In contrast, our method completed the same task in an average of 20 seconds, with a worst-case runtime of 25 seconds.

---

### Official Review · Reviewer_H83n · 2024-11-04

**Soundness:** 3
**Presentation:** 2
**Contribution:** 3
**Rating:** 6
**Confidence:** 3

**Summary:**

This paper introduces a novel algorithm for efficient sampling of strings from regular languages with specific features (e.g., precise occurrence counts of given symbols). The work is motivated by the need to study how specific language features causally impact learnability by neural models. At the core of the algorithm lies a new algebraic structure called the counting semi-ring, which enables tracking occurrence counts of predefined events when sampling from probabilistic finite state automata (PFSA).
The experimental framework consists of sampling random PFSAs and generating sets of strings Y with target features for each PFSA. To evaluate language learnability, both LSTM and Transformer models are trained on these string sets. The evaluation uses KL divergence between the trained models and the original PFSA, along with a decomposed KL metric to assess the impact of interventions on target features.
The empirical results reveal two key findings: (1) Transformers demonstrate superior performance compared to LSTMs in modeling these languages, and (2) the features that indicate language learnability differ between LSTM and Transformer architectures. These findings contribute to understanding of how different neural architectures learn and generalize regular languages.

**Strengths:**

1. The concept of analyzing neural models through properties of regular languages is valuable and brings a novel perspective to the field. This approach has the potential to deepen our understanding of how neural models can capture structured patterns within regular languages.

2. The proposed algorithm for efficiently sampling arbitrary strings with constrained features represents an important methodological advancement. This tool could facilitate more precise and controlled studies of neural models' behavior in relation to regular languages.

3. The design of various features—such as transitions, states, and symbols—along with related explanatory variables, is both creative and effective, as demonstrated in the experiments. This feature framework may be beneficial for further research into neural models and their capabilities with regular languages.

**Weaknesses:**

1. While using a semiring approach to study the counting properties of weighted automata is indeed a classical method in automata theory [1], there is limited comparison between the proposed method and these traditional approaches. Including such a comparison would help clarify the contribution and novelty of this work, positioning it more effectively within the broader research landscape.

2. The use of causality in the current structure appears somewhat extraneous. In this context, causal "interventions" seem equivalent to conditioning rather than introducing true causal effects. Since no causal effects are explicitly evaluated in the experiments, it may improve clarity to describe these interventions simply as conditional operations.

3. Some critical experimental details are missing, which impacts reproducibility and clarity. Specifically, more information on the sampling process for probabilistic finite-state automata (PFSAs) with 100 states and 10 samples would be helpful, such as the random seed selection process and the composition of final sets across different intervention types.

[1] Droste, M. Handbook of Weighted Automata. Springer-Verlag, 2009.

**Questions:**

1. Please refer to the questions in the Weaknesses section.
2. In line 309, the term "DPFSA" is introduced, but prior sections mention only "PFSA". Could the authors clarify the meaning of "DPFSA" and explain how it differs from or relates to "PFSA"?

---

> ### Author Response · Authors · 2024-11-19
> **thanks**
>
> We appreciate the constructive feedback and answer your questions below:
>
> **1. While using a semiring approach to study the counting properties of weighted automata is indeed a classical method in automata theory, there is limited comparison between the proposed method and traditional approaches. A comparison would help clarify the contribution and novelty of this work, positioning it more effectively within the broader research landscape.**
>
> This is a valid point we will address. Reviewer FZWr pointed out the term “counting semiring” has been used before, is this what you refer to? Our “counting semiring” differs from it despite the choice of name, which we will change so there is not a conflict. On the other hand, we now recognize the relation to polynomial semirings, see the discussion with FZWr. We will expand on this to better position our work.
>
> **2. The use of causality in the current structure appears somewhat extraneous. In this context, causal "interventions" seem equivalent to conditioning rather than introducing true causal effects. Since no causal effects are explicitly evaluated in the experiments, it may improve clarity to describe these interventions simply as conditional operations.**
>
> To clarify our point of view: We think of the changes to the generative process as causal interventions on the trained neural networks. This allows us to ask how the model's behavior would change if we modified specific aspects of the language generation process. Even if the do-intervention calculations can sometimes simplify to the conditionals, we still hold that this doesn’t rule out a causal analysis since we are interested in the causal question. We are, however, redrawing the causal graph, the $\Pi$s should not be independent as in the current version, they have a complex joint distribution that leads to backdoor paths, we are fixing this. Could you elaborate on what you mean by “true causal effects”?
>
> **3. More information on the sampling process for probabilistic finite-state automata (PFSAs) with 100 states and 10 samples would be helpful, such as the random seed selection process and the composition of final sets across different intervention types.**
>
> We use the built-in random number generator in Python to sample the transitions between states, along with Dirichlet sampling (numpy.random.dirichlet) to sample the weights so they sum to 1. This is described in lines 338 and down, but we will try and make this clearer in the paper. Is there something specific about the composition of the resulting datasets that you would like to see made clearer?
>
> **4. In line 309, the term "DPFSA" is introduced, prior sections mention only "PFSA". Could you clarify the meaning of "DPFSA"?**
>
> Thanks, we will unify this. DPFSA means a deterministic probabilistic finite automaton, i.e. that a given state has at most a single transition that emits a given symbol. Our experiments only consider DPFSAs, we will clarify this in the writing.

---

> > ### Comment · Reviewer_H83n · 2024-11-20
> > **clarification for question 2**
> >
> > Thank you for your response. I would like to apologize for the ambiguity in my phrasing of question 2. My intention was to make the following points:
> > - The causal interventions in the original graph can be simplified to conditionals.
> > - The experiments do not evaluate any causal effects.
> > If the goal is to answer a causal question, it would be more appropriate to use a formulation explicitly based on causal effects.

---

> > > ### Author Response · Authors · 2024-11-20
> > > **response**
> > >
> > > We want to respond to this in the abstract. Suppose we have a causal graph $G$ over random variable variables $\{X_n\}_{n=1}^N$. Suppose it happens that $p(X_n \mid \mathrm{do} X_m = x) = p(X_n \mid  X_m = x)$. Can you explain why you think this is not a causal effect? I think it's just an edge case where the do calculation happens to coincide with conditioning.
> > >
> > > In our study, we challenge that it follows that because a causal effect reduces to conditioning, it is not a causal effect. They do coincide in some cases.

---

> > > > ### Comment · Reviewer_H83n · 2024-11-20
> > > > **response**
> > > >
> > > > I would like to clarify that the "causal effect" I mentioned there means "averaged causal effects/treatment effects" like $E[Y|do(X=x)]−E[Y|do(X=x')]$. You are right that in some literatures $p(Y=y|do(X=x))$ is called "causal effects". Thank you.

---

> > > > > ### Author Response · Authors · 2024-11-20
> > > > > **response**
> > > > >
> > > > > I understand that we could talk about the average treatment effect. But if the causal graph that induces the distribution we are taking the expectation with respect to just *happens* to have the do-intervention be equivalent to conditioning, is it not fair to call that a causal effect in your eyes?

---

> > > > > > ### Comment · Reviewer_H83n · 2024-11-20
> > > > > > **response**
> > > > > >
> > > > > > No, it does not matter.

---

> > > > > > > ### Author Response · Authors · 2024-11-20
> > > > > > > **response**
> > > > > > >
> > > > > > > Right. Then, in this case, do you think it's fair for us to call it a causal effect then? We are willing to change it if we are confused about how the term is typically used.

---

> > > > > > > > ### Comment · Reviewer_H83n · 2024-11-20
> > > > > > > > **response**
> > > > > > > >
> > > > > > > > If as you said you would revise the causal graph to include some backdoor paths, then it might be necessary to use causality terminologies for analyzing the treatment effects. Besides, for me it would be more clear to refer "P(Y|do(X=x))" as the post-intervention distribution.

---

> > > > > > > > > ### Author Response · Authors · 2024-11-20
> > > > > > > > > **response**
> > > > > > > > >
> > > > > > > > > This is true. The reason we did it this way is that we didn't know, a priori, that the graph would have no backdoor paths. Indeed, we might design one for which our algorithms would be useful that did. The general points in our paper sort of sit at a higher level than one particular graph. And, we would be afraid we would lose the fact that are doing something causal in this case if we just used conditioning notation.

---

> > > > > > > > > > ### Author Response · Authors · 2024-11-28
> > > > > > > > > > **update**
> > > > > > > > > >
> > > > > > > > > > We have updated the pdf with a new causal graph and an additional causal experiment that shows the difference between training on data from intervention sampling and regular ancestral sampling.  It is also visible here https://i.imgur.com/1jfDXPK.png. This means that we treat the automata we sample form as a random variable. We also consider the intervention property and the sampled corpora as jointly distributed without specifying how exactly, acknowledging the complex relation between the two. That is, we intervene on the symbol occurrences and derive a Monte Carlo estimate of the decomposed KL divergence between automata and trained neural models under intervention. The results we get are different from those we see using ancestral sampling if we simply bin the decomposed KL by the natural occurrences of the target symbols.
> > > > > > > > > >
> > > > > > > > > > Do you think this is more along the lines of a true causal effect study?

---

> ### Comment · Reviewer_H83n · 2024-12-01
> **response**
>
> The revised paper improved the clarity in the causality part, which increased my rating. I would like to advise the author to address the question 1 in the next version.

---

> > ### Author Response · Authors · 2024-12-01
> >
> > Thank you!
> >
> > We are currently looking into your first question and would appreciate it if you could clarify what you mean by "classical" approaches for counting properties of automata. We’ve reviewed foundational texts, including Droste, Kuich, and Vogler's Handbook of Weighted Automata, but haven’t found explicit examples of counting properties in the specific sense we address (i.e., tracking and enumerating occurrences of targeted transitions in a probabilistic framework).
> >
> > However, we believe you may be referring to examples like the following, which focus on more global properties of automata or input strings:
> >
> > **Boolean semiring**: In the Boolean semiring, "counting" could be interpreted as detecting whether a string is accepted (1 if accepted, 0 otherwise). While this technically doesn't count transitions, one could sum these values over multiple strings to count how many are accepted.
> >
> > **Natural number semiring**: The natural numbers semiring allows counting paths—for instance, the number of paths that accept a string (by assigning 1 to each transition), or the total number of paths between two states. This aggregates over all paths and counts global properties of the automaton.
> >
> > These examples contrast with our method, which focuses on counting specific transitions during the automaton’s operation. Our semiring allows us to not only sample paths according to their probabilistic weights but also to explicitly encode and count occurrences of a targeted set of transitions within those paths.
> >
> > Additionally, our semiring provides control during sampling. This control mechanism is not inherent in the Boolean or natural numbers semirings, which focus on aggregating properties post hoc rather than influencing path selection during computation.
> >
> > We are happy to include a brief summary of these classical approaches in our work and highlight how our contribution differs by targeting local properties of the automaton and enabling controlled sampling alongside probabilistic modeling.

---

> > > ### Author Response · Authors · 2024-12-03
> > >
> > > We sincerely appreciate the time and effort you have dedicated to reviewing our paper. We wanted to confirm if the examples of the Natural and Boolean semirings align with what you had in mind. Thank you again for your valuable feedback!

---

### Official Review · Reviewer_FMxv · 2024-11-04

**Soundness:** 3
**Presentation:** 3
**Contribution:** 3
**Rating:** 6
**Confidence:** 3

**Summary:**

This paper investigates the ability of LMs to learn formal languages by using causal interventions on the data generation process. Specifically, they introduce a counting semiring that allows them to intervene on the number of transitions (and by extension, states and symbols) in a data-generating automata that occur in a dataset of a fixed size. Their novel methods for sampling from automata improve the efficiency in terms of the maximum count of occurrences to set in the intervention. Through an empirical study, they use their methods to investigate what influences the ability of Transformer and RNN models to learn regular languages.

**Strengths:**

1) A novel sampling method is introduced that could be of independent interest.
2) The paper is well organised and clearly written.
3) The theory is backed by an empirical investigation, showing how the methods could be practically useful.
4) The theory is mathematically rigorous, showing that their construction yields the desired properties.

**Weaknesses:**

1) It seemed insufficiently motivated why intervening on the exact number of transitions would be useful. Could you provide specific examples of potential real-world applications or research scenarios beyond the one explored in the experimental section where this would be valuable?
2) It is unclear how practically impactful the efficiency of the proposed is. Could you provide concrete examples of scenarios where being able to increase the maximum count n further would be particularly beneficial?
3) A more detailed comparison with the existing methods would help contextualise the importance of the efficiency gain. The current version does not make it clear whether the n^4 method mentioned is a simple baseline or the best previously known method.
4) The experimental results in Figure 2 could be better presented. Perhaps additionally including aggregated statistics would be easier to interpret, e.g. plotting the average across the machines with error bars while retaining the current figure in Appendix.

**Questions:**

1) What is the rationale for training RNNs for 4 epochs but Transformers for 10? This appears to make comparisons between the two architectures less clear. Borenstein et al., 2024 from which you adapted the configuration uses two epochs for each architecture. Perhaps results with an equal number of epochs could be included to ensure a fair comparison.
2) Why are the occurrences on the plots at irregular values? Is the specified count for the intervention randomly sampled?
3) How did you choose the specific numbers for the empirical results, e.g. the number of states, symbols, and the counts for interventions?

Nitpicks:
1) Some citations are not formatted well, e.g. parentheses on lines 367 and 968.
2) The figures could be formatted better, e.g. with the values on the x axis being less crowded.

---

> ### Author Response · Authors · 2024-11-18
> **Response**
>
> Thanks for the thorough review!
>
> **1. Further motivation for intervening on the number of transitions:**
>
> We agree that this should be clearer. We are primarily interested in research applications where one can control for the properties of the generated corpus directly at sampling time, both to understand what languages are hard to model, but also how the models generalize:
>
> a) In the SCAN paper (Lake and Baroni, 2017) the authors ask if the models generalize from RUN to JUMP. Using the counting semiring we can sample such settings directly without having to hand curate datasets. The same holds for their length splits, we can simply count all symbols and sample strings of target lengths. Here the benefit is a more efficient sampling algorithm and the fact that we can target the patterns of interest directly by how the automata is constructed, without parsing the output afterwards.
>
> b) Borenstein et al., 2024, consider how the global properties of formal languages impact their learnability. Being able to sample directly based on marked transitions enables us to analyze the effect of the \emph{local structural properties} on the learnability of these languages. We can ask questions such as: what structural properties make it hard to learn this word as we do in our work. That is, the new semiring enables granular controls that allow us to study more complex properties than those based on surface-level properties of the generated strings.
>
> c) We are interested in analyzing generalization from one emitted term (A to B) based on structural similarity. We can directly sample strings to evaluate whether the models are learning to generalize over a given transition in the subgraph. By controlling for the occurrences of B only on the parts of the machine where there is a structural similarity to the parts emitting A, we can ask directly if the model is generalizing from A to B and how much coverage of the shared structure is needed. We don’t see how this could be done another way.
>
> **2. Example where being able to increase the maximum count further is beneficial?**
>
> If we want to do length-controlled sampling as in the length sampling example above. We can then directly sample strings of longer lengths that would take a far longer time to sample with rejection sampling. This is valuable for analyzing e.g. the generalization capabilities from shorter strings to longer strings.
>
> **3. A comparison with existing methods to contextualize the importance of the efficiency gain. Is the n^4 method mentioned the best previously known method?**
>
> It's a simple baseline using the fastest standard algorithms we're aware of. The recent MLRegTest paper (https://arxiv.org/abs/2304.07687) is an example that uses a method similar to the baseline we mentioned, to do length-constrained sampling from a DFA. They intersect a DFA with a DFA for the language $\Sigma^n$, and assign weights using topological sort. This suggests that it runs in $O(n^2)$ time (although they don't provide a runtime analysis), their method is specific to unweighted DFAs and does not implement sampling from the true posterior distribution of a PDFA, which is necessary for our experiments. To modify their algorithm to sample from the posterior, we would need to run weight pushing on each intersected DFA, resulting in $O(n^4)$ time.
>
> **4. The experimental results in Figure 2 could be better presented. Perhaps additionally including aggregated statistics would be easier to interpret, e.g. plotting the average across the machines with error bars while retaining the current figure in Appendix.**
>
> This is a good idea and we will look into updating the figures to better represent the effects.
>
> **5. What is the rationale for training RNNs for 4 epochs but Transformers for 10?**
>
> We found that the RNNs needed a shorter time to converge, while the Transformers kept improving. Our thinking was that instead of trying to match parameters that can impact the two architectures differently we would get the most out of each setting. We should have made this clearer and we will report the number of epochs taken to converge for each architecture.
>
> **6. Why are the occurrences on the plots at irregular values?**
>
> Well caught, we found an issue in our code that led to the values not being fixed except for the symbol interventions, you’ll see they are at fixed intervals. This will be addressed.
>
> **7. How did you choose the specific numbers for the empirical results?**
>
> We choose the highest number of states, symbols, and counts that would fit in memory with a GPU-based implementation. We thought a higher number of states would give rise to more interesting machines. The maximum transition counts chosen exceed those reported for individual symbols as used in Valvoda et al. (2022). In future work, we will use Tarjan’s algorithm for strongly connected components to scale up the analysis to much larger setups.
>
> ----
>
> We’ll also address the citations and figure formatting, thanks!

---

> > ### Comment · Reviewer_FMxv · 2024-11-27
> >
> > Thank you for the response! While I still think some additional motivation for the significance of the efficiency gain could be beneficial, your replies provided useful additional context and I have raised my score to 6.

---

> > > ### Author Response · Authors · 2024-11-27
> > >
> > > Thanks!

---

### Author Response · Authors · 2024-11-28
**Updated manuscript**

Dear Reviewers,

We are sincerely grateful for the extensive feedback and constructive suggestions. We have uploaded a new version, and believe the comments have improved the work significantly. We have focused on improving the presentation and clarifying the causal setting, using a simpler causal model, and providing a more direct causal experiment. We have also added a derivation of the decomposed KL divergence and a proof of the isomorphism between the marginal semiring (formerly counting semiring) and a truncated polynomial semiring.

The main changes are highlighted in blue. We hope these address the main points raised. We are also happy to continue the discussion until the end of the review process.

---

### Meta-Review · Area_Chair_ufrH · 2024-12-23

**Metareview:**

This paper introduces what the authors call a “counting semiring,” which is used to exert fine-grained control over string sampling from probabilistic finite state automata (PFSAs). The motivation is to enable causal interventions on specific string-level features (such as symbol frequencies, occurrences of particular transitions or states) and to use these interventions to study how these features affect language model learnability. The authors present a theoretical framework using this new algebraic construction, implement sampling algorithms, and then apply these to train and evaluate Transformer and LSTM models on synthetic formal languages.

Key Contributions:
	•	A framework for “intervention-based” sampling from PFSA that allows controlling specific properties of strings.
	•	An algebraic construction (initially termed a “counting semiring,” later referred to as a “marginal semiring”) and its associated algorithms for efficient sampling.
	•	Empirical demonstrations on how controlling string-level properties might highlight differences in how Transformers and LSTMs learn from synthetic languages.

**Additional Comments On Reviewer Discussion:**

Positive Points Identified by Reviewers:
	•	The idea of controlling targeted features in sampled data to analyze causal aspects of model learnability is interesting and potentially useful.
	•	The sampling algorithm and its complexity improvements might be of independent interest.
	•	The paper’s direction—using formal languages as a testbed for understanding model limitations—is valuable.

Concerns Raised by Reviewers:
	1.	Motivation and Scope of Causal Claims:
Multiple reviewers noted that the causal framing of the work is not fully substantiated. While the paper uses the “do”-operator and discusses interventions, the experiments often amount to conditioning rather than demonstrating genuine causal effects. The causal graph initially presented was simplified, and while the authors introduced a revised experiment and causal graph, reviewers remained unconvinced that the paper genuinely shows non-trivial causal effects. The interventions are largely identical to conditioning on counts, raising questions about whether this is truly a causal analysis or simply a controlled data generation experiment.
	2.	Practical Impact and Significance:
Reviewers questioned the real-world applicability and significance of the introduced sampling approach. The improvements in sampling efficiency and complexity are demonstrated only in a very narrow synthetic setting. There was a desire for a clearer demonstration of how these methods enable substantially new forms of inquiry that would be difficult or impossible otherwise. Reviewers felt that the experiments—focusing on how changes in occurrences of certain transitions or symbols affect learning—did not yield particularly novel or surprising insights. The observed facts, like Transformers outperforming LSTMs, are known and not directly actionable.
	3.	Clarity of Presentation and Notation:
Several reviewers found the paper difficult to read due to notational complexity, pushing key definitions to the appendix, and inconsistent usage of variables. The theoretical construction, while mathematically sound, could have been presented more straightforwardly, with clearer examples and without overly general formalism that is not used in the experiments.
	4.	Novelty of the Semiring Construction:
Reviewers pointed out that a semiring similar to the proposed “counting semiring” already exists in the literature, at least in concept (e.g., polynomial semirings, truncated polynomial semirings, or known counting constructions). While the authors provided additional context and adjusted their claims, the core theoretical contribution was seen as less groundbreaking than initially presented.
	5.	Experiments and Results:
Although the authors added new experiments and revised parts of the manuscript during the discussion, reviewers generally felt the findings were incremental and not strongly motivating the complexity of the new approach. The causal angle remained tenuous, and the demonstrated efficiency improvements did not convincingly show a must-have tool for future work.

Reviewer Scores and Discussion:
	•	Reviewers were initially split, with some scoring marginally above and others marginally below the acceptance threshold.
	•	The authors addressed some reviewer concerns, improved clarity, and provided additional complexity analyses.
	•	However, the fundamental issues—insufficiently clear causal framing, limited empirical significance, and unclear practical impact—were not fully resolved.
	•	While one or two reviewers increased their scores slightly, the final consensus remained that the paper’s contributions, while interesting, do not rise to a level that justifies acceptance.

Recommendation:
Despite the authors’ efforts to address feedback during the discussion period, the revised submission does not fully dispel the reviewers’ reservations. The paper’s contributions are overshadowed by questions about the depth and genuineness of the causal claims, the real significance of the experimental findings, and the somewhat complicated presentation that makes the main ideas less accessible. At the current stage, the paper would benefit from a more direct demonstration of compelling causal insights, a clearer exposition, and a stronger argument for the practical utility of the proposed method.

---

### Decision · Program_Chairs · 2025-01-22

Reject